


# Measurements of higher alkanes using NO+ PTR-ToF-MS: significant contributions of higher alkanes to secondary organic aerosols in China

Chaomin Wang[1,2],Caihong Wu[1,2],Sihang Wang[1,2], Jipeng Qi[1,2], Baolin Wang[3], Zelong Wang[1,2], Weiwei Hu[4], Wei Chen[4], Chenshuo Ye[5], Wenjie Wang[5], Yele Sun[6], Chen Wang[3], Shan Huang[1,2], Wei Song[4], Xinming Wang[4], Suxia Yang[1,2], Shenyang Zhang[1,2], Wanyun Xu[7], Nan Ma[1,2], Zhanyi Zhang[1,2], Bin Jiang[1,2], Hang Su[8], Yafang Cheng[8], Xuemei Wang[1,2], Min Shao[1,2,*], Bin Yuan [1,2,*]

[1] Institute for Environmental and Climate Research, Jinan University, 511443 Guangzhou, China

[2] Guangdong-Hongkong-Macau Joint Laboratory of Collaborative Innovation for Environmental Quality, 511443 Guangzhou, China

[3] School of Environmental Science and Engineering, Qilu University of Technology, 250353 Jinan, China

[4] State Key Laboratory of Organic Geochemistry and Guangdong Key Laboratory of Environmental Protection and Resources Utilization, Guangzhou Institute of Geochemistry, Chinese Academy of Sciences, 510640 Guangzhou, China

[5] State Joint Key Laboratory of Environmental Simulation and Pollution Control, College of Environmental Sciences and Engineering, Peking University, 100871 Beijing, China

[6] State Key Laboratory of Atmospheric Boundary Physics and Atmospheric Chemistry, Institute of Atmospheric Physics, Chinese Academy of Sciences, 100029 Beijing, China

[7] State Key Laboratory of Severe Weather & Key Laboratory for Atmospheric Chemistry of China Meteorology Administration, Chinese Academy of Meteorological Sciences, 100081 Beijing, China

[8] Multiphase Chemistry Department, Max Planck Institute for Chemistry, Mainz 55128, Germany

*Email: Bin Yuan (byuan@jnu.edu.cn) and Min Shao (mshao@pku.edu.cn)

**Abstract:** Higher alkanes are a major class of intermediate-volatility organic compounds (IVOCs), which have been proposed to be important precursors of secondary organic





aerosols (SOA) in the atmosphere. Accurate estimation of SOA from higher alkanes and
their oxidation processes in the atmosphere are limited, partially due to difficulty in their
measurements. High-time resolution (10 s) measurements of higher alkanes were performed
using a novel online mass spectrometry method at an urban site of Guangzhou in Pearl River
Delta (PRD) and at a rural site in North China Plain (NCP), respectively. High concentrations
were observed in both environments, with significant diurnal variations. At both sites, SOA
production from higher alkanes is estimated from their photochemical losses and SOA yields.
Higher alkanes account for significant fractions of SOA formation at the two sites, with
average contributions of 7.0±8.0% in Guangzhou and 7.1±9.5% in NCP, which are
comparable or even higher than both single-ring aromatics and naphthalenes. The significant
contributions of higher alkanes in SOA formation suggests that they should be explicitly
included in current models for SOA formation. Our work also highlights the importance of
$NO^+$ PTR-ToF-MS in measuring higher alkanes and quantifying their contributions to SOA
formation.



## 1. Introduction

As important components of fine particles, secondary organic aerosols (SOA) not only affect air quality and climate change, but also threaten human health (An et al., 2019;Zhu et al., 2017;Chowdhury et al., 2018). Recent studies indicate large discrepancies between simulations and observations for SOA (de Gouw et al., 2008;Dzepina et al., 2009;Jiang et al., 2012), which are attributed to limited understanding of complicated chemical and physical processes underlying SOA formation (Hallquist et al., 2009). A volatility basis set (VBS) model was developed to advance SOA modeling by lumping numerous, yet unidentified, precursors based on their volatility (Donahue et al., 2006), which substantially improved the agreement between SOA simulations and observations (Hodzic et al., 2010). However, there are still large uncertainties in current VBS models, including rate constants of oxidation reactions, the change of O/C ratio in oxidation, and the relative importance of functionalization and fragmentation (Ma et al., 2017;Hayes et al., 2015). Explicit consideration of individual or a group of important semi-volatile or intermediate volatile organic compounds (S/I-VOCs) in the SOA model are urgently needed.

Higher alkanes as a major class of IVOCs (roughly corresponding to alkanes with 12-20 carbons) have been proposed as important SOA contributors in urban areas (Robinson et al., 2007;Yuan et al., 2013;Zhao et al., 2014a). Higher alkanes are estimated to produce as much as or even more SOA than single-ring aromatics and polycyclic aromatic hydrocarbons from the oxidation of  vehicle emissions, based on the chemical compositions measurements of vehicle exhausts (Tkacik et al., 2012b;Zhao et al., 2016a). Previous model studies suggested that SOA simulation can be significantly improved when higher alkanes were considered in the model (Pye and Pouliot, 2012;Jathar et al., 2014). Although the concentrations of higher alkanes might be lower than other VOCs classes (e.g. aromatics) in



the atmosphere, higher alkanes are found to have much higher SOA yields and the yields
increase steadily with carbon number (Lim and Ziemann, 2005;Lim and Ziemann,
2009;Presto et al., 2010a). For a given carbon number, SOA yields of higher alkanes reduce
with branching of the carbon chain, especially under high-$NO_x$ conditions (Lim and Ziemann,
2009;Tkacik et al., 2012a;Loza et al., 2014).

Higher alkanes, predominantly *n*-alkanes, have been mainly measured by gas

chromatography-based techniques, focusing on the compositions (Gong et al., 2011;Caumo
et al., 2018), atmospheric concentration levels (Bi et al., 2003;Anh, 2018) and gas-particle
partitioning (Xie et al., 2014;Sangiorgi et al., 2014). While most of previous studies collected
offline samples (usually 0.5 day-1 week) for GC-based analysis in the laboratory, hourly
online measurements of *n*-alkanes using GC-based thermal desorption aerosol gas
chromatograph for semi-volatile organic compounds (SV-TAG) was recently developed and
applied in ambient air (Zhao et al., 2013). Proton-transfer-reaction mass spectrometry (PTR-
MS) using $H_3O^+$ as reagent ions are capable of measurements for many organic compounds
with high time response and sensitivity (de Gouw and Warneke, 2007;Jordan et al.,
2009;Yuan et al., 2017b). Although $H_3O^+$ PTR-MS is response to large alkanes (>C8), these
alkanes usually fragment into small masses with mass spectra difficult to interpret (Jobson
et al., 2005;Gueneron et al., 2015). Recently, PTR-MS using $NO^+$ as reagent ions was
demonstrated to provide fast online measurement of higher alkanes (Erickson et al.,
2014;Koss et al., 2016;Inomata et al., 2013). The high-time resolution measurements of
higher alkanes provide valuable information for SOA estimation, as the dependence of SOA
yields on organic aerosol concentrations and other environmental parameters (e.g.
temperature) (Lim and Ziemann, 2009;Presto et al., 2010a;Loza et al., 2014;Lamkaddam et
al., 2017a) can be taken into account in more details.





In this study, we utilize NO$^+$ PTR-ToF-MS to measure higher alkanes at two different
sites in China, one urban site in Pearl River Delta region and one rural site in North China
Plain region. We use the datasets along with measurements of other pollutants to estimate
contributions to SOA formation from higher alkanes and other SOA precursors. The
observation-constrained SOA formation of this study is a step forward upon previous
modelling studies, which proposed the important roles of S/I-VOCs (Jiang et al., 2012;Yang
et al., 2018;Wu et al., 2019) including higher alkanes(Yuan et al., 2013) in SOA formation
in China.

## 2.  Methods

Field campaigns were conducted at an urban site of Guangzhou in the Pearl River
Delta (PRD) region during September-November 2018 and at a rural site of Baoding in North
China Plain (NCP) during November-December 2018, respectively. The detailed description
of the measurement sites can be found in Supporting Information (SI, Figure S1).

## 2.1 NO$^+$ PTR-ToF-MS measurements

Proton-transfer-reaction mass spectrometry (PTR-MS) is a technique that allows for
fast and sensitive measurements of volatile organic compounds (VOCs) at trace levels in air.
PTR-MS using H$_3$O$^+$ chemistry makes possible of quantitative of alkenes, aromatics, and even
oxygenated VOCs (Yuan et al., 2017a). Here, PTR-MS with NO$^+$ chemistry was used to detect
higher alkanes, through hydride abstraction by NO$^+$ forming mass (m-1) ions (m is the
molecular mass) (Erickson et al., 2014;Koss et al., 2016;Inomata et al., 2013).
A commercially available PTR-ToF-MS instrument (Ionicon Analytik, Austria) with
a mass resolution of 5500 m/$\Delta$m was used for this work. To generate NO$^+$ as reagent ions,
ultra-high- purity air (5.0 sccm) was directed into the hollow cathode discharge ion sources.



The pressure of the drift tube was maintained at 3.8 mbar. Voltages of ion source and drift
chamber were explored (Figure S2) to optimize the generation of $NO^+$ ions relative to $H_3O^+$,
$O_2^+$, and $NO_2^+$ and minimize alkane fragmentation. Ion source voltages of Us and Uso were
selected as 40 V and 120 V, while Udrift and Udx were set to 470 V and 23.5 V, resulting in
an E/N (electric potential intensity relative to gas number density of 60 Td. $NO^+$ PTR-ToF-
MS data was analysed using Tofware software (Tofwerk AG) for high-resolution peak-fitting.
A description of the algorithm can be found in Stark et al. (2015) and Timonen et al. (2016).
Figure 1 shows the high-resolution peak fitting to the averaged mass spectra on a typical day
(12 October 2018) for m/z 169, m/z 211 and m/z 281, at which masses produced by dodecane
($C_{12}H_{25}^+$), pentadecane ($C_{15}H_{31}^+$) and eicosane ($C_{20}H_{41}^+$) are detected. It is observed that the
ions from higher alkanes lie at the right-most position at each nominal mass, with signals
either the largest or among the largest ions at these nominal masses, which help to achieve
high precision for determined signals of higher alkanes from high-resolution peak fitting
(Cubison and Jimenez, 2015;C. Corbin et al., 2015).

In this study, we normalize the raw ion count rate of higher alkanes to the primary ion

($NO^+$) at a level of $10^6$ cps to account for fluctuations of ion source and detector. Calibrations
were conducted every 1-2 days under both dry conditions (RH<1%) and ambient humidity
conditions using a gas standard with a series of *n*-alkanes (Apel Riemer Environmental Inc.)
during both two field campaigns (Figure 2(a)). Calibration factors of *n*-alkanes (C8-C15)
standards were stable during the campaigns (Figure S3). Humidity-dependent behaviours of
these *n*-alkanes (C8-C15) were performed in the laboratory under different humidity by
diluting higher alkanes standard into humidified air to reach approximately 1 ppb mixing ratio.
As shown in Figure 2(b, c), like *n*-dodecane and *n*-pentadecane, the normalized signal of all
higher alkanes decreased slightly with increasing humidity. The humidity effects on higher
alkanes for both campaigns were corrected using the laboratory established relationships.



The fragmentation patterns for selected *n*-alkanes and their branched isomers are
measured with $NO^+$ PTR-ToF-MS by introducing commercially acquired pure chemicals
(Figure S4). Figure 3(a) shows the fractions of hydride abstraction m-1 ions in the mass
spectra of C8-C20 *n*-alkanes in $NO^+$ PTR-ToF-MS. Generally, larger *n*-alkanes show less
degree of fragmentation in the mass spectra with higher fractions contributed by m-1 ions.
The fraction of m-1 ions account for more than 60% of total ion signals for >C12 *n*-alkanes.
We also observe good correlation between the fractions of m-1 ions in mass spectra and the
determined sensitivities for C8-C15 *n*-alkanes. As C16-C21 *n*-alkanes exhibit similar degrees
of fragmentation as C15, sensitivities of the alkanes were assumed to be same as that of C15
*n*-alkane (Figure 3(b)). Comparison of the degree of fragmentation between *n*-alkanes and
their branched isomers (Figure S5) show the substituted group affect little on the degrees of
fragmentation for product ions, at least for branched isomers with up to 4 substituted methyl
groups. Previous studies demonstrated that the branched alkanes from emissions of fossil fuel-
related sources are primarily with one or two alkyl branches (Chan et al., 2013;Isaacman et
al., 2012). Therefore, we conclude that the branched isomers of higher alkanes should have
similar response factors to their normal analogues. As a result, the concentration of higher
alkanes by $NO^+$ PTR-ToF-MS should be regarded as the summed concentrations of *n*-alkanes
and branched alkanes that have the same chemical formulas.
Detection limits are calculated as the concentrations at which signal counts are 3 times
of standard deviation of measured background counts (Bertram et al., 2011;Yuan et al., 2017b).
As shown in Table 1, detection limits are determined to be on the order of 0.7-1.3 ppt for
higher alkanes for 1 min integration times. Response time is calculated as the time required
to observe a $1/e^2$-signal decay after quick removal of the analyte from the sampled air
(Mikoviny et al., 2010). Response times for various alkanes are better than 1 min, expect for
C21 alkanes (116 s) (Table1).



During these two campaigns, PTR-ToF-MS automatically switches between $H_3O^+$ and
$NO^+$ chemistry every 10-20 minutes with a 10 s resolution of measurement. Ambient air was
continuously introduced into PTR-ToF-MS through a Teflon tubing with an external pump at
5.0 L/min. The calculated residence time in Teflon tubing is ~7.6 s for PRD campaign and
~3.0 s for NCP campaign, respectively. PTR-ToF-MS instrument was operating in a room
with a constant temperature of 20 ℃. An insulating tube with stable temperature of 40 ℃ was
used to wrap outside of the sampling tubing to avoid water vapor condensation. Background
measurement of 3 minutes was conducted in each cycle of $NO^+$ and $H_3O^+$ measurements by
introducing the ambient air into a catalytic converter with a constant temperature of 367 ℃.
**2.2 Other measurements**
During the Guangzhou campaign, an online GC-MS/FID system was used to measure
C2-C11 alkanes, alkenes and aromatics with a time resolution of one hour (Yuan et al., 2012).
Non-refractory components in particulate matter with diameter less than 1μm ($PM_1$)
including nitrate, sulfate, ammonium, chloride, and organics were measured with an
Aerodyne high-resolution time-of-flight aerosol mass spectrometric (HR-ToF-AMS) and a
time-of-flight aerosol chemical speciation monitor (ToF-ACSM) in PRD and NCP,
respectively. Trace gaseous species (CO, NO, $NO_2$, $O_3$, and $SO_2$) were measured using
commercial gas analyzers (Thermo Scientific). Photolysis frequencies were measured using
a spectroradiometer (PFS-100, Focused Photonics Inc.). In addition, temperature, pressure,
relative humidity and wind were continuously measured during two campaigns.
**3.  Results and Discussion**
**3.1    Ambient concentrations and diurnal variations of higher alkanes**





184 Although NO$^+$ chemistry has been shown to be valuable in measuring many organic

185 species, the applications in real atmosphere of different environments are still limited(Koss et

186 al., 2016). Here, we compared the measurements of various VOCs from NO$^+$ PTR-ToF-MS

187 with both H$_3$O$^+$ PTR-ToF-MS and GC-MS/FID during the two campaigns. Overall, good

188 agreements between PTR-ToF-MS (both H$_3$O$^+$ and NO$^+$ chemistry) and GC-MS/FID are

189 obtained for aromatics (Figure S6). Consistent results of oxygenated VOCs are also obtained

190 between H$_3$O$^+$ and NO$^+$ chemistry (Figure S7). The time series and diurnal variations of

191 alkanes (C8-C11) between NO$^+$ PTR-ToF-MS and GC-MS/FID are shown in Figure 4 (and

192 Figure S8). Similar temporal trends for these alkanes are observed from the two instruments.

193 However, the concentrations at each carbon number from NO$^+$ PTR-ToF-MS are ~3-6 times

194 those from GC-MS/FID. This is expected, as dozens to hundreds of isomers exists for alkanes

195 with carbon number at this range (Goldstein and Galbally, 2007) and GC-MS/FID only

196 measured one or a few isomers. Based on measurements from NO$^+$ PTR-ToF-MS and GC-

197 MS/FID, we compute the molar concentration fractions of *n*-alkanes for each carbon number

198 (Figure 5). We found the fractions are in the range of 11%-21% for carbon number of 8-11,

199 which are comparable with results of ambient air in California, vehicle exhausts and diesel

200 fuel (Figure 5) (Chan et al., 2013;Gentner et al., 2012;Isaacman et al., 2012). These results

201 indicate the importance of branched alkanes in concentrations of higher alkanes and their

202 potential contributions to SOA formation. It also has strong implication for the merits of NO$^+$

203 PTR-ToF-MS in measuring sum of the alkanes with the same formula for estimation of SOA

204 contributions, as discussed later.

205 Table 2 summarizes means and standard deviations of concentrations of C8-C21

206 higher alkanes measured in PRD and in NCP, respectively. The mean concentrations of *n*-

207 alkanes measured at a suburban site in Paris (Ait-Helal et al., 2014) and an urban site in

208 Pasadena, U.S. (Zhao et al., 2014a) are also included in Table 1 for comparison. According





to the fraction of *n*-alkanes, the mean concentrations of *n*-alkanes in China are found to be
comparable to that from Paris and higher than in Pasadena. In general, concentrations of
higher alkanes concentration decrease with the increase of carbon number, with octanes (C8)
at ~0.5 ppb and heneicosanes (C21) at ~0.002 ppb. This decrease pattern of carbon distribution
are as the results of lower emissions from sources (Gentner et al., 2012), larger reactivity
towards OH radicals (Atkinson et al., 2008;Keyte et al., 2013) and larger fractions partitioning
to particles (Liang et al., 1997;Xie et al., 2014;Zhao et al., 2013) in the atmosphere.

The diurnal variations of selected higher alkanes are shown in Figure 6. C12 alkanes

and C15 alkanes exhibit similarly strong diurnal variations at both sites, with a relatively
high levels at night and minimum concentrations detected in the late afternoon at both sites.
Such diurnal variations are consistent with other primary VOCs species (e.g. aromatics), as
the result of both shallow boundary layer heights at night and faster chemical removal in the
daytime. The diurnal profiles of other higher alkanes are similar to C12 and C15 alkanes.
**3.2    Estimation of the contributions of higher alkanes to SOA formation**

A time-resolved approach based on consideration of photooxidation processes with

OH radical (Ait-Helal et al., 2014) was applied to estimate contributions of higher alkanes
to SOA during these two campaigns. In order to evaluate the relative importance to SOA
from different precursors, the same method was also used for monoaromatics, naphthalenes,
and isoprenoids.

This method considers the amount of chemical removal based on the parameterized

photochemical age, which was widely used to quantify contributions of different VOC
precursors to SOA formation (Zhao et al., 2014a;Ait-Helal et al., 2014;de Gouw et al., 2009).
The contributions to SOA formation from different compounds are determined by the ratios
of calculated SOA production amounts from individual precursors and SOA concentrations





derived from factor analysis of OA measurements by AMS (SI, Appendix 3). In this method,
SOA formation for a given compound can be estimated by
$$[SOA_i]_t = [VOC_i]_t \times (e^{k_{VOC_i} \times ([OH] \times \Delta t)} - 1) \times Yield_i \qquad\qquad (1)$$

where $[SOA_i]_t$ is the calculated SOA production (μg m$^{-3}$) for a given specific compound $VOC_i$
at time $t$, $[VOC_i]_t$ is the $VOC_i$ concentration measured at time $t$ (μg m$^{-3}$), $Yield_i$ is the SOA
yield data summarized from chamber studies, $k_{VOC_i}$ is the rate constant of $VOC_i$ with the OH
radical (cm$^3$ molecule$^{-1}$ s$^{-1}$). The OH exposure, $[OH] \times \Delta t$ (molecules cm$^{-3}$ s), is estimated by
the ratio m+p-xylene and ethylbenzene with different reactivity for anthropogenic VOCs and
by the oxidation processes of isoprene for biogenic VOCs, respectively (Apel et al., 2002) (see
details in Figure S9). Since biogenic emissions were pretty weak during cold winter (mean
temperature 0.5±3.6 °C) during NCP campaign, measured concentrations of isoprene and
monoterpenes are attributed to be of anthropogenic origin during the winter campaign in NCP
campaign, especially given the fact that they showed similar variations, diurnal profiles and
strong correlation with CO and anthropogenic VOCs species (Figure S10). A previous study
in Helsinki also found the importance of anthropogenic emission in monoterpene
concentrations (Hellén et al., 2012).
Based on equation (1), SOA production from higher alkanes (C8-C21 alkanes),
monoaromatics (benzene, toluene, C8 aromatics, C9 aromatics, styrene), naphthalenes
(naphthalene, methylnaphthalenes, dimethylnaphthalenes) and isoprenoids (isoprene,
monoterpenes) were calculated. The OH reaction rate constant of each compound was taken
from literature (Atkinson, 2003). SOA yield data used here for higher alkanes (Lim and
Ziemann, 2009;Presto et al., 2010b;Loza et al., 2014;Lamkaddam et al., 2017b),
monoaromatics (Li et al., 2016;Ng et al., 2007;Tajuelo et al., 2019), naphthalenes (Chan et al.,
2009) and isoprenoids (Ahlberg et al., 2017;Carlton et al., 2009;Edney et al., 2005;Kleindienst





et al., 2006;Pandis et al., 1991) were summarized from reported values in the literature, with
the consideration of the influence of organic aerosol concentration (Figure S11) to SOA yields
(Donahue et al., 2006) (Figure S12-13).

Both OH reaction rate constants and SOA yields of *n*-alkanes reported in the literature

are applied for higher alkanes, as most of the chamber studies have focused on n-alkanes. The
OH reaction rate constants of branched alkanes are higher than those of n-alkanes, while their
SOA yields are lower than n-alkanes, of which both depending on chemical structures of the
carbon backbone (Lim and Ziemann, 2009;Tkacik et al., 2012a;Loza et al., 2014). The
combined effects are hard to evaluate due to limited data, but the two effects can partially (if
not all) cancel out. As shown above, temperature (mean temperature 0.5±3.6 °C) in NCP
winter campaign was significantly lower than the temperature (usually 25 °C) at which SOA
yields are derived from chamber studies. Temperature can significantly influence SOA yields,
with higher yields at lower temperature (Takekawa et al., 2003;Lamkaddam et al., 2017b). It
might cause underestimation of SOA production from various precursors in winter of NCP.

The calculated results of SOA production for different higher alkanes are shown in

Figure 7. Although lower concentrations of heavier alkanes were observed for both
campaigns, the calculated SOA production are largest for C12-C18 (Figure 7(b)). This is
because of two reasons: (1) Alkanes with larger carbon number have larger SOA yields. The
calculated SOA yields using the average OA concentrations during the two campaigns are
both larger than 0.2 for >C12 alkanes and increase to near unity for C20-C21 alkanes. (2)
Larger alkanes are relatively more reactive than lighter ones, which results in larger
proportions of calculated concentrations that have been chemically consumed in the
atmosphere. It is interesting to note that the distribution of contributions from alkanes with
different carbon number to SOA formation shown here is in good agreement with the
previous results referred from volatility calculation for precursors (de Gouw et al.,


2011;Liggio et al., 2016). The magnitudes of photochemical processes are apparently
different between the two campaigns, with larger calculated OH exposure in the PRD
campaign in autumn than the NCP campaign in winter (Figure S9). Consequently, the
calculated chemical losses of alkane concentrations and their SOA production are much
higher in PRD, though measured alkane concentrations are comparable during the two
campaigns.

Along with higher alkanes, SOA production for monoaromatics, naphthalenes and

isoprenoids are shown in Figure 8 (and Figure S14-16). The total average SOA production
from C8-C21 alkanes are $0.6\pm0.8$ µg m$^{-3}$ and $1.1\pm1.2$ µg m$^{-3}$ in PRD and NCP, respectively.
The formed SOA from higher alkanes account for $7.0\pm8.0\%$ and $7.1\pm9.5\%$ of SOA formation
in PRD and NCP, respectively. The contributions of monoaromatics to SOA formation are
$6.2\pm7.7\%$ and $9.4\pm17.4\%$ in PRD and NCP, respectively. Naphthalenes have been proposed
to be important precursors of SOA from laboratory chamber studies (Kleindienst et al., 2012).
In this study, we determine $2.8\pm4.6\%$ of SOA in PRD and $11.1\pm14.3\%$ of SOA in NCP are
contributed by naphthalenes. The SOA contribution from naphthalenes determined for NCP
is comparable to the results ($10.2\pm1.0\%$) obtained during haze events in Beijing in a recent
study (Huang et al., 2019). Significant contribution from monoterpenes to SOA ($8.7\pm14.6\%$)
is observed in NCP. As mentioned above, we attribute these isoprene and monoterpenes to
anthropogenic emissions, including vehicle exhausts and biomass combustions in this region.
The SOA precursors considered in this study in total could explain 14.9%-29.0% and 18.1-
119.9% of SOA formation in PRD and NCP, respectively. The lower explained percentages
of SOA formation during the highly polluted periods and during the daytime in NCP (Figure
S14(b)) imply that some other SOA precursors or formation pathways (e.g. aqueous reactions)
are contributing significantly to SOA formation of the strong haze pollution in NCP.
Compared to a previous study in northern China (Yuan et al., 2013), the missing gap of SOA





formation declined after explicitly considering higher alkanes and naphthalenes in SOA
production.

As shown in Figure 8, we find that C8-C21 higher alkanes contribute significantly to

SOA formation at both an urban site in autumn of PRD and a rural site in winter of NCP.
The contributions from higher alkanes are either comparable or higher than both
monoaromatics and naphthalenes. An independent estimation method by considering SOA
instantaneous production rates obtained similar results (Figure S18), which confirms the
results from the photochemical age based on parameterization method shown above. The
importance of higher alkanes in SOA formation has been also proposed in several previous
SOA modelling studies(Pye and Pouliot, 2012;Zhao et al., 2014b). These results, along with
our results from observations in ambient atmosphere, underline that the inclusion of higher
alkanes in SOA models in the atmosphere should be considered if possible.
**4.  Conclusions**

In this study, we utilized a NO$^+$ PTR-TOF-MS to measure C8-C21 alkanes in two

different environments in China. Based on a series of laboratory experiments, we show that
NO$^+$ PTR-TOF-MS can provide online measurements of higher alkanes with high accuracy
and fast response. The measured concentrations of higher alkanes were relatively high during
the two campaigns. The diurnal profiles of higher alkanes are similar to anthropogenic VOCs,
implying they are emitted from anthropogenic sources.

On the basis of measurements of higher alkanes by NO$^+$ PTR-TOF-MS, we successfully

take into account their contributions in SOA formation. The time-resolved measurements of
higher alkanes by NO$^+$ PTR-ToF-MS provide the opportunity to accurately apply the
photochemical age-based parametrization method. As there is no separation before detection
in PTR-ToF-MS, the measured concentrations of NO$^+$ PTR-ToF-MS represent all of the





compounds that contribute to the product ions (m-1 ions), which include concentrations from
both *n*-alkanes and branched alkanes. With the total concentration of both *n*-alkanes and
branched alkanes quantified, the contribution from higher alkanes at each carbon number can
be estimated as a whole. This is an important supplementary method to the traditional
analytical method by GC techniques for higher alkanes, as fully chemical separation and
detection of numerous isomers of higher alkanes remain as a challenge, even using the most
advanced GC×GC-ToF-MS instruments (Chan et al., 2013;Alam et al., 2016).
Higher alkanes were found to have significant contributions to SOA in both PRD and
NCP regions with a similar or even higher contributions than that of monoaromatics and
naphthalenes. The importance of higher alkanes to SOA formation also call for more work
to investigate emissions and chemistry of these compounds in the atmosphere. It was shown
that fossil-related combustions such as vehicle exhausts are major sources for higher alkanes
(Zhao et al., 2016b). While, recent studies pointed out the potential large influence of non-
conbustion sources, e.g., solvent use, on emissions of higher alkanes (McDonald et al.,
2018;Khare and Gentner, 2018). However, such quantitative information on emissions of
higher alkanes is still limited. The measurements of higher alkanes by $NO^+$ PTR-ToF-MS
with fast response could help to fill these research gaps.



## Acknowledgements


This work was supported by the National Key R&D Plan of China (grant No.
2018YFC0213904, 2016YFC0202206), the National Natural Science Foundation of China
(grant No. 41877302), Guangdong Natural Science Funds for Distinguished Young Scholar
(grant No. 2018B030306037), Guangdong Provincial Key R&D Plan (grant No.
2019B110206001), Guangdong Soft Science Research Program (grant No.
2019B101001005) and Guangdong Innovative and Entrepreneurial Research Team Program
(grant No. 2016ZT06N263). Weiwei Hu and Wei Chen were supported by National Natural
Science Foundation of China (grant No. 41875156). The authors gratefully acknowledge the
science team for their technical support and discussions during the campaigns in PRD and
NCP.

## Data availability

Data is available from the authors upon request

## Competing interests

The authors declare that they have no conflicts of interest

## Author contributions

BY and MS designed the research. CMW, CHW, SHW, JPQ, BLW, WC, CW, WS and
WYX contributed to data collection. CMW performed the data analysis, with contributions
from ZLW, WWH and CSY. CMW and BY prepared the manuscript with contributions from
other authors. All the authors reviewed the manuscript.



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





**Table 1**. Fractions of product ions (m-1) ions in mass spectra, sensitivities, response time
and detection limits of higher alkanes in NO+ PTR-ToF-MS.

| Compounds | Ions | Fractions of (m-1) ions (%) | Sensitivities (ncps/ppb) | Response time (s) | Detection limit for 10 s integration (ppt) | Detection limit for 1 min integration (ppt) |
|---|---|---|---|---|---|---|
| *n*-Octane | $C_8H_{17}^+$ | 24 | 104.6 | 9.0 | 3.5 | 1.3 |
| *n*-Nonane | $C_9H_{19}^+$ | 32 | 106.3 | 13.3 | 3.2 | 1.2 |
| *n*- *n*-Decane | $C_{10}H_{21}^+$ | 39 | 120.9 | 14.1 | 3.5 | 1.3 |
| *n*-Undecane | $C_{11}H_{23}^+$ | 44 | 140.9 | 4.2 | 3.3 | 1.2 |
| *n*-Dodecane | $C_{12}H_{25}^+$ | 62 | 156.3 | 2.0 | 2.4 | 0.9 |
| *n*-Tridecane | $C_{13}H_{27}^+$ | 61 | 186.6 | 3.4 | 2.1 | 0.8 |
| *n*-Tetradecane | $C_{14}H_{29}^+$ | 64 | 220.7 | 18.2 | 1.9 | 0.7 |
| *n*-Pentadecane | $C_{15}H_{31}^+$ | 84 | 205.5 | 7.6 | 1.7 | 0.6 |
| *n*-Hexadecane | $C_{16}H_{33}^+$ | 95 | / | 20.0 | 1.6 | 0.6 |
| *n*-Heptadecane | $C_{17}H_{35}^+$ | 82 | / | 30.7 | 1.8 | 0.7 |
| *n*-Octadecane | $C_{18}H_{37}^+$ | 90 | / | 34.9 | 1.8 | 0.7 |
| *n*-Nonadecane | $C_{19}H_{39}^+$ | 71 | / | 28.8 | 1.2 | 0.4 |
| *n*-Eicosane | $C_{20}H_{41}^+$ | 86 | / | 53.4 | 1.9 | 0.7 |
| *n*-Heneicosane | $C_{21}H_{43}^+$ | / | / | 115.9 | 2.0 | 0.7 |







**Table 2.** Mean concentrations of alkanes (C8-C21) in different sites worldwide.

| Compounds | Formula | PRD, China[a] Alkanes (ppt) | PRD, China[b] n-Alkanes (ppt) | NCP, China[a] Alkanes (ppt) | Paris, France[c] n-Alkanes (ppt) | Pasadena, US[d] Alkanes (ppt) |
|---|---|---|---|---|---|---|
| Octane | $C_8H_{18}$ | 482±488 | 50±49 | 463±329 | / | / |
| Nonane | $C_9H_{20}$ | 208±186 | 43±32 | 319±228 | 14±13 | / |
| Decane | $C_{10}H_{22}$ | 174±199 | 29±28 | 307±222 | 24±22 | / |
| Undecane | $C_{11}H_{24}$ | 129±138 | 21±17 | 239±194 | 19±16 | / |
| Dodecane | $C_{12}H_{26}$ | 122±120 | / | 156±119 | 22±21 | 8±1 |
| Tridecane | $C_{13}H_{28}$ | 66±60 | / | 109±75 | 13±12 | 6±1 |
| Tetradecane | $C_{14}H_{30}$ | 50±47 | / | 60±40 | 27±23 | 9±2 |
| Pentadecane | $C_{15}H_{32}$ | 45±42 | / | 42±27 | 23±18 | 5±0.8 |
| Hexadecane | $C_{16}H_{34}$ | 36±33 | / | 27±17 | 22±19 | 4±1 |
| Heptadecane | $C_{17}H_{36}$ | 21±20 | / | 16±11 | / | 3±0.4 |
| Octadecane | $C_{18}H_{38}$ | 13±14 | / | 10±8 | / | 1.6±0.5 |
| Nonadecane | $C_{19}H_{40}$ | 5±9 | / | 3±6 | / | 0.7±0.2 |
| Eicosane | $C_{20}H_{42}$ | 0.7±4 | / | 2±5 | / | 0.24±0.08 |
| Heneicosane | $C_{21}H_{44}$ | 0.5±5 | / | 2±4 | / | 0.15±0.1 |

[a]: alkanes measured with NO[+] PTR-ToF-MS; [b]: n-alkanes measured with GC-MS; [c]: n-alkanes from Ait-Helal
et al. (2014); [d]: n-alkanes from Zhao et al. (2014a).




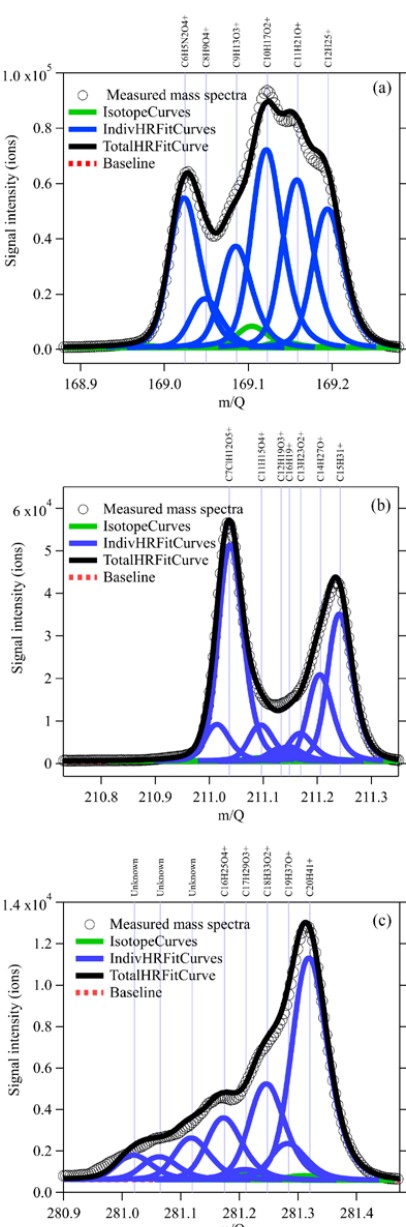


**Figure 1.** High-resolution (HR) peak-fitting to the averaged mass spectra on a typical day (12

October 2018) for m/z 169 **(a)**, m/z 211 **(b)** and m/z 281 **(c)**, at which masses produced by

dodecane ($C_{12}H_{25}^+$), pentadecane ($C_{15}H_{31}^+$) and eicosane ($C_{20}H_{41}^+$) produced in $NO^+$ PTR-ToF-

MS.





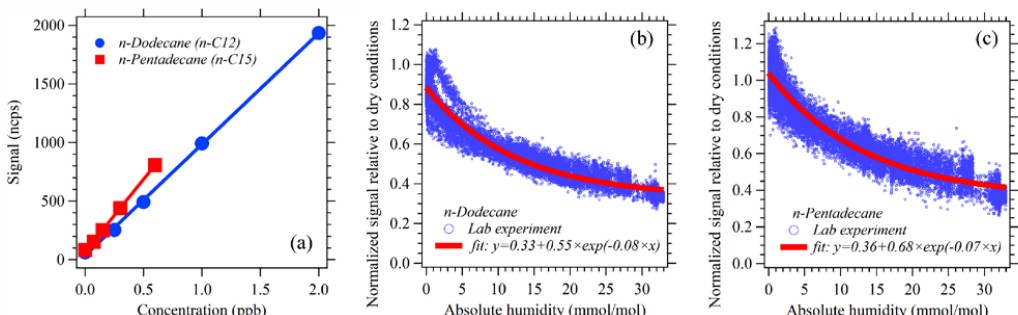


**Figure 2**. **(a)** Calibrations of n-Dodecane and n-Pentadecane under dry conditions; **(b)**
Humidity dependence of n-Dodecane. **(c)** Humidity dependence of n-Pentadecane.





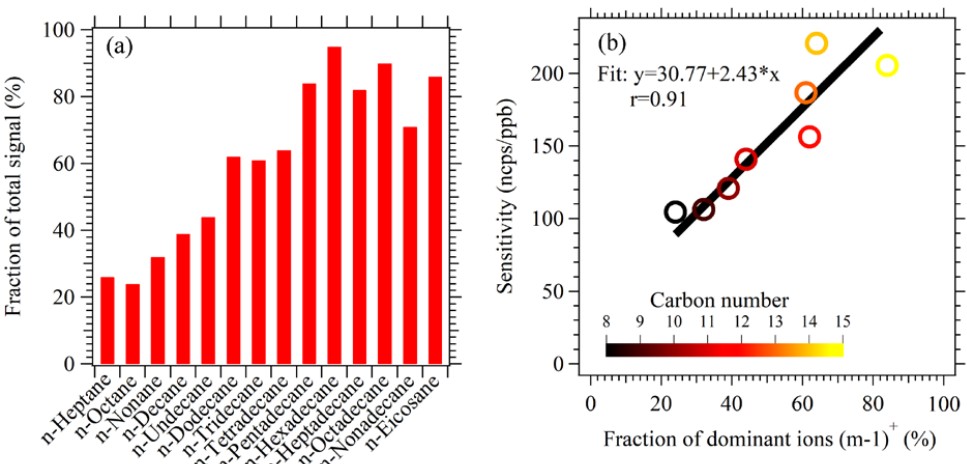


**Figure 3. (a)** The fracitons of product ions (m-1) from hydride abstraction of C8-C20 *n*-alkanes

in NO$^+$ PTR-ToF-MS. **(b)** Scatterplot of sensitivities under dry conditions versus the fractions

of hydride abstraction ions for C8-C15 *n*-alkanes.




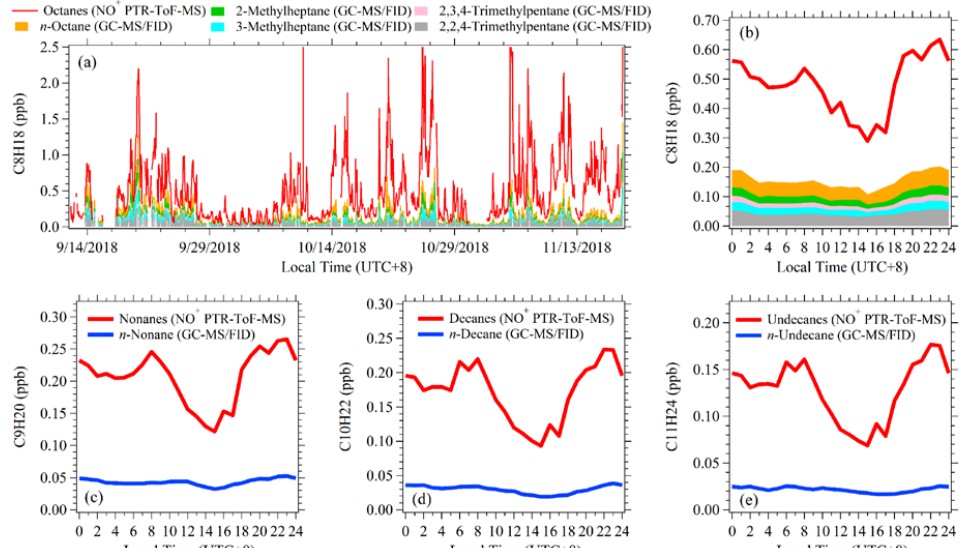

**Figure 4.** Comparisons of times series and diurnal variations of alkanes measured by NO+ PTR-ToF-MS and GC-MS/FID in PRD. **(a)** Time series of C8 alkanes measured by NO+ PTR-ToF-MS, C8 *n*-alkane and four branched isomers measured by GC-MS/FID. **(b)** Diurnal variations of C8 alkanes. **(c-e)** Diurnal variations of C9-C11 alkanes with NO+ PTR-ToF-MS and C9-C11 *n*-alkanes with GC-MS/FID.



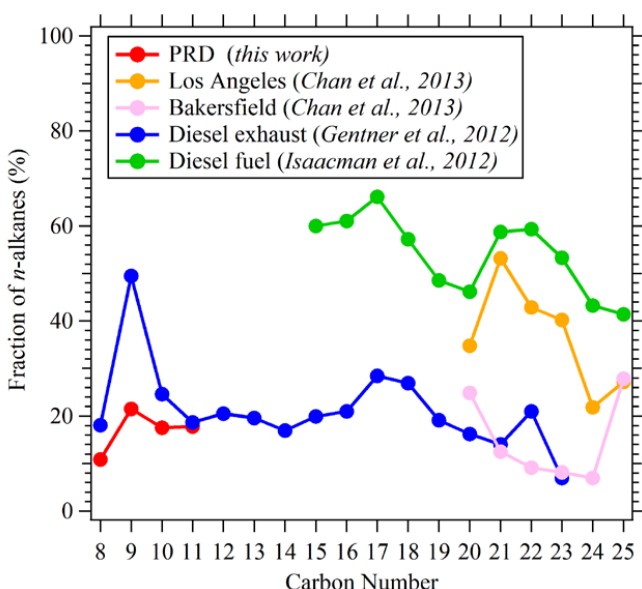


**Figure 5.** Fractions of *n*-alkanes in higher alkanes derived from this study, ambient air in

Los Angeles, Bakersfield and in diesel fuel/exhausts (Chan et al., 2013;Gentner et al.,

2012;Isaacman et al., 2012).






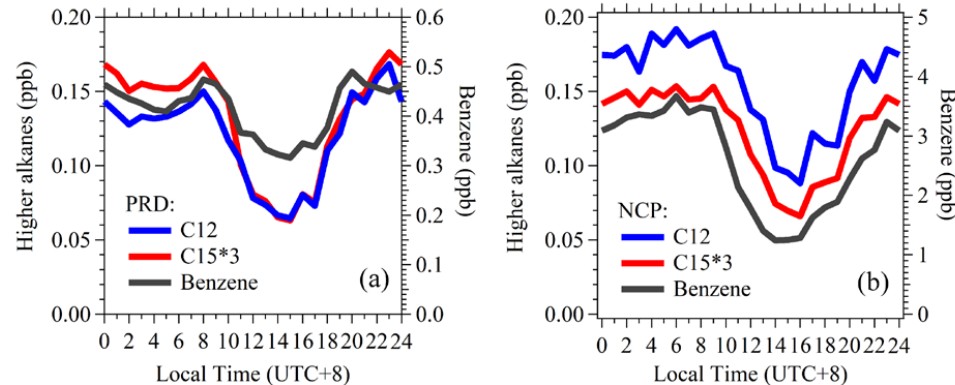


**Figure 6.** Diurnal variations of C12 alkanes, C15 alkanes and benzene in PRD **(a)** and NCP
**(b)**.

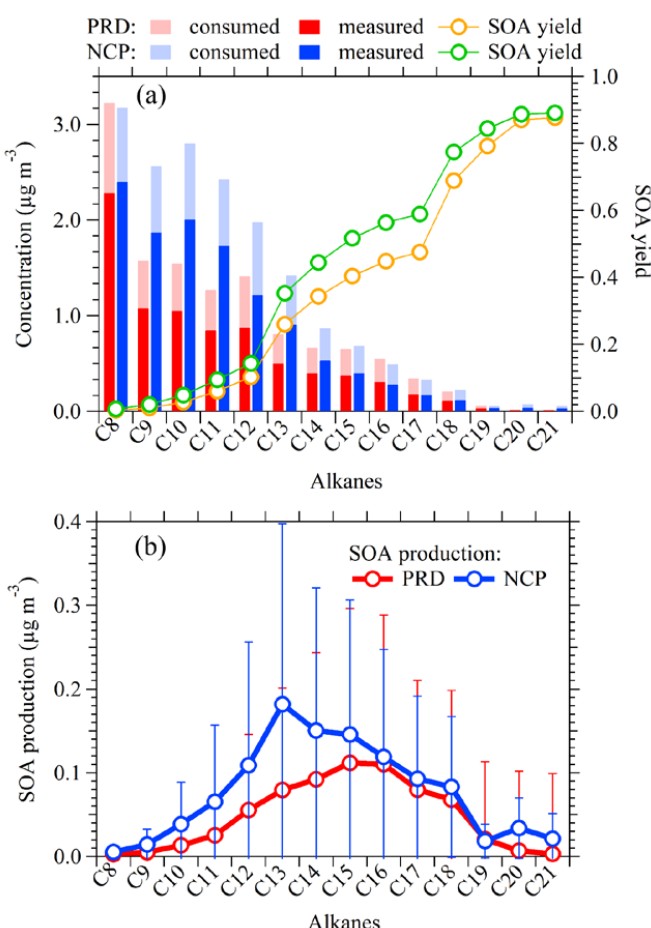


**Figure 7. (a)** Measured concentrations by NO$^+$ PTR-ToF-MS, calculated consumed

concentrations and average SOA yields for C8-C21 alkanes in PRD and NCP. **(b)** Calculated

average SOA productions for C8-C21 alkanes in PRD and NCP.


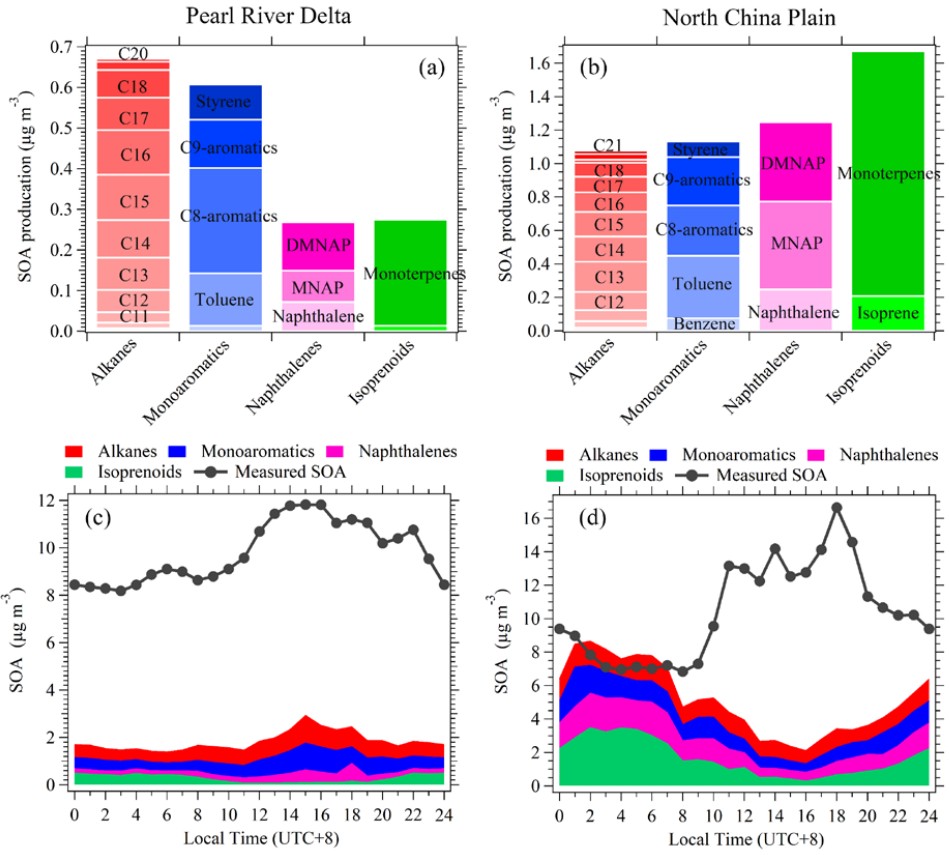


**Figure 8.** The mean concentrations of SOA produced from higher alkanes (C8-C21 alkanes),

monoaromatics (benzene, toluene, C8 aromatics, C9 aromatics and styrene), naphthalenes

(naphthalene, methylnaphthalenes, dimethylnaphthalenes) and isoprenoids (isoprene and

monoterpenes) in PRD **(a)** and NCP **(b)**. Diurnal variations of SOA production from higher

alkanes, monoaromatics, naphthalenes and isoprenoids as well as the measured SOA

concentrations in PRD **(c)** and NCP **(d)**.