# Peer review of "Measurements of higher alkanes using NO+ PTR-ToF-MS: significant contributions of higher alkanes to secondary organic aerosols in China"

_Atmospheric Chemistry and Physics, 2020_

## Referee Comment (RC1) · Anonymous Referee #1 · 21 Apr 2020

General comments:

This paper describes the measurement of higher alkanes by NO+ PTR-ToF-MS at two sites in China and the authors showed the significant contribution of the higher alkanes to secondary organic aerosol (SOA) formation. I recognize that the measurement of the higher alkanes in ambient air is very challenging, so the data presented here would be precious. But I feel that there are two issues in the present paper.

Firstly, the authors seemed to fail to suppress the formation of O2+ in the ion source (Figure S2). I think that the O2+ reaction could substantially interfere with the ion signals of alkanes. In addition, I feel that the ratio of O2+ to NO+ was not stable as

far as I looked at Figure S2. The authors should pay attention to the intensity of O2+ relative to the NO+ intensity.

Secondly, I feel that the diurnal variation of the OH exposure derived from isoprene chemistry method is strange (Figure S9). I cannot understand why the value is low during daytime compared with that in nighttime. The authors indicated two reference (Apel et al., 2002; Carlton et al., 2009), but I could not find the derivation of the OH exposure by "the isoprene chemistry method" in the references. The authors should show how the OH exposure by the isoprene chemistry method was calculated. I think that the low OH exposure during the daytime causes the low SOA formation from isoprenoids during the daytime, as shown in Figure 8(c).

Therefore, the paper is not acceptable until the two issues are resolved. Specific and technical comments are listed below.

Specific comments:

(1) Page 6, Line 113: In Figure S2, the authors showed that the relative intensity of O2+ to that of NO+ was $\sim$50 % (Fig. S2(a)) and $\sim$15 % (Fig. S2(b)) under the conditions of Us = 40 V and Uso = 120V. These conditions are not good because I think that the O2+ reaction could substantially interfere with the ion signals of alkanes. I am concerned that the ratio of O2+ to NO+ was not stable. The authors should mention the quality of the data of the alkanes presented in this paper.

(2) Page 6, Lines 130-131: I cannot agree to the argument by the authors that calibration factors were stable during the campaigns, because the normalized sensitivities of the n-C15 alkane were scattered with a factor of 2. As far as I experienced, the normalized sensitivity is very stable. I feel that this instability of the normalized sensitivity may be related to the interference of the O2+ reaction.

(3) Page 7, Lines 137-139: When I looked at the mass spectra in Figure S4, I found that the signal intensities of 13C-isotopologue of [M-1]+ are low. For example, the

signal at m/z 282 should be 20 % of the intensity at m/z 281 for n-Eicosane. I wonder if the quantitativity was guaranteed or not in the present measurements using the NO+ PTR-ToF-MS instrument.

(4) Page 9, Lines 187-190: Some are good, but some are not good. The authors should mention the results correctly and explain the disagreement for some species. Were the concentrations from PTR-ToF-MS in Figs. S6 and S7 calibrated or calculated?

(5) Page 10, Lines 219-221: The degree of the chemical removal of hydrocarbons by the OH reaction was quite different at two sites, when we consider the concentration of OH (Figure S19). The authors should mention the results accurately.

(6) Page 10, Line 226: The data of naphthalenes were not shown anywhere in this paper.

(7) Page 11, Lines 239-242: I could not understand how the authors estimated the OH exposure clearly. Which value was used as the ratio of m+p-xylene to ethylbenzene at t = 0 (the emission ratio)? How was the OH exposure estimated by the oxidation process of isoprene? Why was the OH exposure estimated by the oxidation process of isoprene low during daytime compared with that in nighttime?

(8) Page 13, Lines 288-89: The diurnal variation of the estimated SOA produced from isoprenoids shown in Figure 8(c) is strange to me. In addition, I cannot understand the diurnal variations of the calculated SOA produced from higher alkanes, monoaromatics, naphthalenes, and isoprenoids in Figure 8(d). Why were those concentrations low during the daytime in spite of the fact that the production rates were calculated to be positive during the daytime (Figure S18(d)). Did the authors consider the dynamics (e.g., the movement of the boundary layer) into the calculation? If so, expalin in the text.

(9) Page 32, Figure 8(d): The diurnal variation of the measured SOA in Figure 8(d) seems to be different from that in Figure S11(b). Is it OK?

(10) Page S5, Line 106: Explain "isoprene chemistry method".

(11) Page S7, Line 129: Which equation was used in the present paper, eqn. (1) in Page 11, Line 235 or eqn. (7) in Page S7, Line 129? Don't make readers confused.

Technical comments:

(1) Page 4, Line 72: Anh et al., 2008. ("et al." is missing)

(2) Page 4, Line 83-84: Erickson et al. (2014) did not use NO+ as the reagent ion. Don't make readers misread. Same comment to Page 5, Line 108.

(3) Page 6, Line 125: Corbin et al., 2015. (Delete "C.")

(4) Page 9, Line 208: Table 1 → Table 2

(5) Page 14, Line 319-Page 15, Line 347: I feel it strange that some papers are cited in "Conclusions". If the authors want to cite the papers, I think that the authors change "Conclusions" to "Concluding remarks".

(6) Page 17, Line 404: "C. Corbin, J." → "Corbin, J. C."

(7) Page S6, Line 107: Figure S8 → Figure S9

(8) Page S7, Lines 134-143: These are mentioned in the text (Page 11, Line 249-Page 12, Line 259)

(9) Page S17, Figure S9: "expousre" → "exposure"

---

## Referee Comment (RC2) · Anonymous Referee #2 · 11 May 2020

Wang et al measured concentrations of VOC and IVOC alkanes in two locations in China. They use the measurement data to estimate the importance of alkanes to ambient SOA. Based on their estimates, these alkanes make up ∼7% of ambient SOA in both sampling locations in north and south China.

Overall the manuscript is topically relevant to ACP. It presents what seems to be a novel application of chemical ionization MS for estimating SOA production. However, I have several comments before I can recommend publication.

Major comments:

(1) I have an issue with calling the measurement technique NO+ PTR-MS. The "P" in

[Figure]

PTR stands for proton, which in this case you are switching out for NO+ ions. So the measurement technique is chemical ionization with NO+ or selected ion mass spectrometry, but not PTR. An exception would perhaps be if you're using PTR-MS to indicate the instrument itself (e.g., an Ionicon PTR-MS) that has been modified.

(2) Line 162-170 introduce that the PTR was run alternately with NO+ and H3O+ every 10-20 minutes. This cycling requires more explanation. How was the cycling achieved? How long did the system take to re-equilibrate when the water flow was turned on and off? Even if the system only took 1-2 minutes to re-equilibrate after switching between ions, a large fraction of the data would be lost. Were data during the transition period considered for analysis? Were voltages and pressures changed or held constant in the NO+ and H3O+ operating modes?

(3) This paper relies on predicted SOA production to compare the importance of alkanes versus aromatics and other compound classes. However the SOA estimates rely on a number of assumptions (e.g., SOA yields) that are uncertain. There are other ways to compare the importance of alkanes to other compounds. There is very little discussion of absolute concentrations, which is one way to make the comparison. The authors could also compare something like OH reactivity.

(4) The comparisons in Figure 8 (and associated discussion) do not seem to place alkanes and aromatics on equal footing. The alkane signal is essentially the sum of all C_x alkanes (i.e., C12 is the sum of dodecane, cyclododecane, and all branched isomers). This is analogous to integrating individual peaks in the GC along with the entire C12 "bin" of UCM (unresolved complex mixture) as shown by Zhao et al (2016a, b) and others. However, the authors only consider specific aromatics (e.g., BTEX) but not the aromatic UCM of larger substituted benzenes. A more complete comparison of alkanes to aromatics would include these species as well.

Minor comments:

(1) Lines 131-136 and Figure 2 - The figure would be easier to interpret if the relative

humidity was also shown.

(2) Line 192, in the comparison of GC-MS and NO+ ionization for alkanes: "Similar temporal trends for these alkanes are observed from the two instruments." I can't really tell this from the figure. The diurnal trends suggest that the PTR-derived measurements have a deeper afternoon trough than the GC-MS, which seems to show a flatter concentration profile.

(3) Line 193-194 - PTR data are a factor of 3-6 higher than the GC data. It looks like the GC data were only analyzed for species that are resolved by the GC (e.g., no UCM). How does the closure look if the entire chromatogram (or the entire m/z 57 signal) is binned and analyzed, similar to Zhao et al's (2016a, b) work?

(4) Why does Figure 5 only show PRD data for carbon numbers 8-11? The instrument measured up to C21.

Grammar:

(1) Line 80 states that PTR "is response to large alkanes" - please edit. I think the authors mean that PTR responds to large alkanes.

(2) Line 160 - replace expect with except
* * *

---

## Referee Comment (RC3) · Anonymous Referee #3 · 15 May 2020

Comments on acp-2020-145

In their manuscript, "Measurements of higher alkanes using NO+ PTR-ToF-MS: significant contributions of higher alkanes to secondary organic aerosols in China," the authors tackle an important but often overlooked issue: the contribution of high-molecular-weight gas-phase alkanes to SOA formation. A relatively large body of work over the last decade has provide a fairly thorough examination of the composition of vehicle emissions and the importance of branched (and cyclic) alkanes; this manuscript uses this work as a starting place to explore their contribution to SOA in China. The work is made possible in large part by the recent development of the NO+ PTR. Overall,

I think this work addresses interesting questions, and makes real efforts to tackle the complexity of calibrating a new instrument for compound classes that may contain hundreds of isomers. However, there are a few scientific weakspots that need to be addressed before this work is ready for publication. These larger comments include some needs for clarification, and some more fundamental issues with how OH exposure and SOA potential are being calculated. I commend the authors for taking a step into some unknown territory, but some additional discussion (and possibly work) needs to be included.

General comments:

1) There are some typo and english issues throughout. It is not sufficient to seriously detract from the science, but it is to the degree that the authors should have a native english speaker review and edit this manuscript.

2) Considering the focus of this work is on large alkanes, and the semi-volatile nature of these components, particularly at the lower temperatures of some of these measurements, this manuscript really does not provide enough detail on the sample line and inlet. How long is this sample line? Is it heated all the way to the sample inlet, or just in the room? If it is not heated all the way to the inlet, I would have some misgivings about alkanes about C14 or so, there could be substantial losses or time lags for larger alkanes. Is there any evidence (observations or models) that suggest line losses and delays can be ignored? If the authors want to focus on gas-phase measurements of S/IVOC alkanes, more detail needs to be provided for the sampling.

3) In the discussion around "sensitivity", is that the response per mass of the m-1 ion, or the sum of all ions? If the former, it should be discussed in some of the relationships shown, as discussed in comments below. If the latter, how are those ions summed and attributed in the complex real-world data? Understanding of this terminology is critical for interpreting some of the figures.

4) More detail is necessary on how they calculate OH exposure. I'm not familiar with

the isoprene chemistry method - please include a description. And for the ratio-of-aromatics method, don't you need to start by assuming some ratio at the time of emissions? What is assumed here? Why does the OH exposure diurnal (Figure S9) look very different for the biogenics and the anthropogenics? Notably, in the NCP campaign this is also a large missing daytime source, could this be related to a "mistiming" of SOA caused by a bias in the OH exposure calculation? Or what might be the missing source?

The math on Eq. 1 makes sense, but I'm not sure the assumptions do. This assumes that the air behaves like a plug flow reactor from some emissions source, and then just oxidizes until the measurement site. Wouldn't local emissions (that perhaps have a different oxidation history) change things dramatically? Given that benzene and TMB aren't present in diesel or motor oil (which is the source of higher alkanes), this might be an issue. Though they are both from vehicle emissions, so maybe not. This might all be more compelling with more detail on how OH exposure is calculated.

Specific comments:

Line 135. The relationship between humidity and sensitivity seems fairly robust, so correctable, but I'm not sure a would classify a 50% drop in sensitivity as a "slight" decrease. If the "sensitivity" is to m-1, is the drop due to a change in fragmentation, or a drop in all ions? As it also due to a shift of NO+ to H3O+ as water content increases, or does increasing humidity not increase H3O+?

Line 144. Is this slope between fraction of m-1 and sensitivity just due to the decrease m-1? In other words, is the total ion count produced the same, and just the fragmentation different, or is the total ion count lower for lower alkanes?

Line 144-145. Are calibrations of C16-C21 not available directly just because it is hard to get them into the gas phase, or was there some other reason?

Line 160. Should be "except" instead of "expect"

Line 162. How was this switching achieved? Was it just a change in voltages? Are there residual effects observed, and/or do some transitional data need to be removed?

Line 167. I'm not quite sure what the "insulated tube" is - do they mean heater rope or something?

Line 185. Do the authors mean there are limitations to its application? Or just that it hasn't been applied very often?

Line 187-190. The comparison between H3O+ and NO+ and GC-MS/FID is very encouraging. There are some significantly non-unity slopes though (0.47 for benzene) - do the authors have an explanation for this?

Line 276. A Table of SOA yields used would be helpful for the SI

Line 301-304. Are there emissions sources for only alkanes? I would guess not, in which case these alkanes must be getting emitted alongside cyclic saturated hydrocarbons (e.g., cycloalkanes), which actually contribute similar or more to petroleum fuels (e.g., Gentner et al., 2012, Table S5). These compounds are expected to have broadly similar (or slightly higher) SOA yields. This would have you substantially overestimating SOA in NCP at night - the authors should comment on possible explanations or biases.

Figure 5. I'm not quite sure what data is being shown here. Gentner provides diesel fuel data in table S5 of that publication (this is used to approximate exhaust in that manuscript). That data doesn't agree with what is shown here though - for C8 it is 37%, and all the values are near or above 21%. The Isaacman paper is actually just a single fuel sample that is included in the Gentner data set, so could probable be excluded. That paper also provides gasoline data, which is not shown. In Figure 9 of the Chan paper, the branched-to-normal ratios are given for vehicle exhaust, which convert to between 13 and 41% normal for C24 through C21, which are fairly different than those shown. Similarly, in Figure 5, the branched-to-normal ratios for LA are shown at

multiple oxidation times, and reach levels of 20 to 50% for those same carbon numbers - this is simlar to the values shown, but I think not quite right. I think the numbers for Bakersfield (based on Figure 4 from that paper) should be much lower, 5-10%, for all carbon numbers. Values should also be included for direct vehicle emissions from Worton et al., 2014, dx.doi.org/10.1021/es405375j. Basically, some explanation for how these numbers were all compiled is necessary (even if it's just in the SI), because they don't look quite right to me.

---

## Referee Comment (RC4) · Anonymous Referee #4 · 29 May 2020

General comments:

This work is a nice, extensive manuscript using an NO+ CIMS (see below) to measure ambient alkanes in the PRD of China and evaluate their importance to SOA chemistry. The experimental methods are high quality and very well-documented. The importance of this manuscript is high, and I find it to be novel and useful.

It is, however, difficult to understand in places and suffers from some lack of precision in language. I recommend publication after these relatively minor, but common issues are fixed.

Specific comments:

[Figure]

Instrument name: The title of the paper and description of the instrument is not correct. PTR is a widely used term that refers to the transfer of a proton from H3O+ under controlled conditions. NO+ ionization uses different ionization mechanisms, particularly charge transfer. I understand the instrument is a commercial PTR-MS, but the authors are now using a different reagent ion and the authors should change the name to something like NO+ ToF-MS or NO+ CIMS. Analytical chemistry acronyms are confusing enough as it is. Removing all meaning from them will make them unintelligible to everyone.

SOA Yields: There would appear to be a large amount of possible error in these measurements and calculations and it needs to be reflected when yields (SOA production) are displayed across the paper. Some yield calculations are presented with errors and some are not. Figure 7 and 8, in particular have issues with this.

P2 L29: Is it novel? This method has previously been published (with GC comparisons) by some of the co-authors of this paper in Koss et al.

P12 L280: Is this really a surprise? Why? Do those references use different yields or inputs than this work?

Figure 1: The mass-to-charge labels on the top of the graph are entirely illegible and the labels on the axes are too small to be conveniently legible. Please make larger. I understand that the authors used the default Tofware labels, but "IndivHRfitCurves" will not mean a lot to most readers. Please change the labels to something clearer in each graph.

Figure 5 gives the impression that the authors suggest the dominant source of alkanes in the region is diesel vehicles. Is that correct? If not, I'm not sure I understand the point of putting those traces on the graph with ambient measurements.

Figure 7a needs error bars on the calculated SOA yields. There would appear to be a large amount of possible error in these measurements and calculations and it needs to

be reflected when yields are displayed in this work

Technical corrections:

P4 L80: should be "is responsive" P4 L82: Not PTR-MS. See above. P4 L88: "in more detail." P5 L102: See above. Not PTR-MS P5 L110: "mass resolving power" instead of resolution P5 L111: "source" P15 L343: "While..." this sentence needs to be re-written for grammatical clarity

Many small language clarity issues throughout the manuscript.

References:

Koss, A. R. et al. Evaluation of NO+ reagent ion chemistry for online measurements of atmospheric volatile organic compounds. Atmos. Meas. Tech. 9, 2909–2925 (2016).

---

## Referee Comment (RC5) · Anonymous Referee #5 · 31 May 2020

Wang et al. report on measurements of higher alkanes made with a PTR-ToF-MS in an urban and rural location in China. They find relatively abundant quantities of higher alkanes and, based on estimates of OH concentrations and SOA mass yields, argue that they also contribute meaningfully to SOA formation in both regions. Measuring the chemical composition and complexity of VOCs in the atmosphere is an important step in studying the potential of those VOCs to form SOA and identify the sources of fine particulate matter. The study is well motivated and mostly well written (some detailed comment about technical communication can be found below). I am not overtly familiar with the online mass spectrometry measurements so the editor should rely on the other reviews to make a judgement about that. The data analysis and modeling sections

require some additional detail to better communicate the inputs and assumptions. I generally favor publication of this work in ACP but after the authors have had a chance to respond to my comments.

Comments:

1. Line 2: After reviewing the manuscript, I did not find that the word 'significant' was appropriate in the title. The authors find that alkanes are probably as important as aromatics and biogenic VOCs but the model currently still underestimates the total SOA production in both studied regions. The word 'significant' could be misconstrued to mean that alkanes explain the majority of the SOA in urban and rural china.

2. Lines 47-53: This introduction to SOA modeling is not well described and does accurately represent the historical approaches used to model SOA formation. For instance, precursor lumping has been used prior to the volatility basis set.

3. Lines 58-61: It might be worthwhile to talk about the differences in the various SOA precursor classes between gasoline and diesel exhaust.

4. Lines 61-63, lines 314-316: A recent study by Akherati et al. (ACP, 2019) also modeled SOA formation from higher alkanes but did not find them to be as important when compared to aromatics, especially after accounting for the effects of vapor wall losses (see #12).

5. Line 80: 'is response' should be 'responds'.

6. Line 105: grammatical mistake in 'makes possible of quantitative of alkenes'.

7. Lines 147-148: grammatical mistake in 'group affect little on the degrees of fragmentation for product ions'.

8. Lines 193-195: I don't see why the PTR measurement should be higher than the GC. Shouldn't the GC measurement be close to the PTR if the unresolved mixture is accounted for?

[Figure]

9. Line 212: Should be 'This decreasing pattern'.

10. Lines 239-242: It isn't clear to me why different pairs of species are used to estimate OH concentrations for the anthropogenic and biogenic species separately. Can the equation for the OH estimation be provided? Also, is there confounding in the OH estimation associated with emissions being added to the air parcel while the selected pairs are oxidized?

11. Lines 242-246: If the isoprene and monoterpenes are of anthropogenic origin, shouldn't the biogenic VOCs be oxidized based on the OH determined from the anthropogenic VOCs?

12. Lines 253-onwards: While I commend the authors for relying on a lot of historical data to determine SOA parameterizations for the VOCs measured in this work, the toluene parameterizations do not use the more recent work of Zhang et al. (PNAS, 2014) that account for the influence of vapor wall losses in chambers. On a related note, were the SOA mass yields adjusted for vapor wall losses? Alternatively, some of the parameterizations can also be found in Eluri et al. (ACP, 2018). Finally, how was the NOx-dependence on SOA modeled? No NOx data at either of those sites was presented, nor any argument was made for the use of high NOx SOA parameters.

13. Lines 299-300: The sentence reads as if isoprene and monoterpenes are emitted in vehicle exhaust. Please correct.

14. Line 312: Can you describe how this is 'independent'?

15. Line 332: The previous literature that has examined alkanes in more detail (Gentner et al., 2012; Zhao et al., 2015,2016; Drozd et al., 2019) have found a strong contribution of cyclic alkanes, especially in vehicular exhaust. Were these specifically measured/estimated in this work and can the authors comment on their role in contributing to SOA formation?

---

## Author Comment (AC1) · 20 Aug 2020

**Response to anonymous referee #1**

**2 General comments:**

1

This paper describes the measurement of higher alkanes by NO+ PTR-ToF-MS at two sites in China and the authors showed the significant contribution of the higher alkanes to secondary organic aerosol (SOA) formation. I recognize that the measurement of the higher alkanes in ambient air is very challenging, so the data presented here would be precious. But I feel that there are two issues in the present paper.

8 Response: Thank you very much for your valuable comments and suggestions on our
9 manuscript. These comments are all helpful for improving our article. All the authors have
10 seriously discussed about all these comments. According to your comments, we have tried best
11 to modify our manuscript to meet with the requirements for the publication in this journal.
12 Point-by-point responses to the comments are listed below.

Firstly, the authors seemed to fail to suppress the formation of  $O_2^+$  in the ion source (Figure S2). I think that the  $O_2^+$  reaction could substantially interfere with the ion signals of alkanes. In addition, I feel that the ratio of  $O_2^+$  to  $NO^+$  was not stable as far as I looked at Figure S2. The authors should pay attention to the intensity of  $O_2^+$ relative to the NO+ intensity.

Response: In the submitted manuscript, we made a mistake on the Fig. S2, where we put a 18 wrong number of Us=40 V (in fact Us=120 V) on the Fig. S2(a). I think that might mislead 19 you into thinking that we faild to suppress the formation of  $O_2^+$ . The corrected data and more 20 21 experiment results were included in the revised Fig. S2. Before field campaigns, we did laboratory tests on the ion source voltages to find suitable volatage settings, where intensities 22 of ion impurites  $(O_2^+, H_3O^+ \text{ and } NO_2^+)$  are low. According to the laboratory results, we chose 23 Us=40 V and Uso=100 V as ion source voltage settings for the field measurement of higher 24 alkanes (Fig. S2(a)), under which condition (absolute humidity=~11.6 mmol/mol) the ratio of 25  $O_2^+$  to NO+ is ~10%. 26

We also included the data of NO+,  $O_2^+$ ,  $H_3O^+$  and  $NO_2^+$  during field campaigns in the revised supporting information. The intensities of primary ion NO+ and impurites ( $O_2^+$ ,  $H_3O^+$ and  $NO_2^+$ ) (**Fig. S3**) and the ratio of  $O_2^+$  to NO+ (**Fig. S4**) fluctuate during two campaigns. The ratio of  $O_2^+/NO^+$  (**Fig. S4(a**)) is basically stable at 2-4% during the PRD campaign except a little high values of 7-10% for Oct. 26-Nov. 2, 2018. While, for the NCP campaign, the ratio of  $O_2^+/NO^+$  (**Fig. S4(b**)) fluctuates between 10-40% in the early stage of campaign and keeps stable at ~20% in the late stage of campaign. When comparing the  $O_2^+/NO^+$  ratio with the ambient abosolute humidity during campaigns (**Fig. S5**), we find that there was an obvious negative correlation between  $O_2^+/NO^+$  ratio and ambient abosolute humidity (**Fig. S5**), which is also consistent with laboratory humidity experiments. As the result, the fluctuations of  $O_2^+/NO^+$  ratios are attributed to the changes of humidity during the two campaigns.

We agree with the comment that  $O_2^+$  could interfere with the ion signals of alkanes. 38 According to the study from Amador et al. (2016), the reactions of  $O_2^+$  with alkanes can be 39 proceeded by charge transfer and hydride abstraction that might affect the ion signals of alkanes 40 with NO+ reactions in our study. As we got the alkanes standard at the very late period of the 41 PRD campaign, we did not have the daily calbirations for this campaign. However, the 42  $O_2^+/NO^+$  ratios were small with a genaral range of 2-4% (Fig. S4(a)), we think the interfence 43 of  $O_2^+$  in this campaign is negligible. During the NCP camaign, we conducted calibrations 44 every 1-2 days under both dry conditions and ambient humidity conditions. In the revised 45 manuscript, we add a figure (Fig. 2) to show the correlation of sensitivities of *n*-alkanes and 46  $O_2^+/NO^+$  ratios. As showned in Fig. 2, the sensitivities of *n*-alkanes (C8-C15) varied 47 significantly with the fluctuations of  $O_2^+/NO^+$  ratios in both dry conditions and ambient 48 conditions during the NCP campaign. This indicate that the interference of  $O_2^+$  should be 49 considered. Therefore, we use the daily ambient calibrations results to quantify the 50 51 concentration of higher alkanes for the NCP campaign.

52

---

## Author Comment (AC2) · 20 Aug 2020

**Response to anonymous referee #2**

Wang et al measured concentrations of VOC and IVOC alkanes in two locations in China. They use the measurement data to estimate the importance of alkanes to ambient SOA. Based on their estimates, these alkanes make up ~7% of ambient SOA in both sampling locations in north and south China. Overall the manuscript is topically relevant to ACP. It presents what seems to be a novel application of chemical ionization MS for estimating SOA production. However, I have several comments before I can recommend publication.

Response: Thank you very much for your valuable comments and suggestions on our manuscript. These comments are all helpful for improving our article. All the authors have seriously discussed about all these comments. According to your comments, we have tried best to modify our manuscript to meet with the requirements for the publication in this journal. Point-by-point responses to the comments are listed below.

**Major Comments:**

(1) I have an issue with calling the measurement technique $NO^+$ PTR-MS. The "P" in PTR stands for proton, which in this case you are switching out for $NO^+$ ions. So, the measurement technique is chemical ionization with $NO^+$ or selected ion mass spectrometry, but not PTR. An exception would perhaps be if you're using PTR-MS to indicate the instrument itself (e.g., an Ionicon PTR-MS) that has been modified.

Response: We understand the reviewer's concern. We use the $NO^+$ PTR-TOF-MS as the instrument name to reflect the fact that $NO^+$ chemistry has been mainly adopted from PTR-MS instruments as a switchable reagent ion technique (Jordan et al. 2009), which is also the case for this work. The first attempt to measure higher alkanes using $NO^+$ chemistry (Inomata et al., 2013) referred this method as $NO^+$ chemical ionization using proton-transfer-reaction mass spectrometry, which seems to reflect the reality of both the ion chemistry and also the instrumentation used for the method. As the result, we changed the title in the revised manuscript: "**Measurements of higher alkanes using $NO^+$ chemical ionization in PTR-ToF-MS: important contributions of higher alkanes to secondary organic aerosols in China**".

In the main text (line 98-99) of the revised manuscript, we also reflect this information:

**"In this study, we utilize $NO^+$ chemical ionization in PTR-ToF-MS (here referred as $NO^+$ PTR-TOF-MS) to measure higher alkanes at two different sites in China."**

(2) Line 162-170 introduce that the PTR was run alternately with $NO^+$ and $H_3O^+$ every

10-20 minutes. This cycling requires more explanation. How was the cycling achieved?

How long did the system take to re-equilibrate when the water flow was turned on and off? Even if the system only took 1-2 minutes to re-equilibrate after switching between ions, a large fraction of the data would be lost. Were data during the transition period considered for analysis? Were voltages and pressures changed or held constant in the

$NO^+$ and $H_3O^+$ operating modes?

Response: We included more details about the automatic switches between $H_3O^+$ chemistry and $NO^+$ chemistry in the revised paper. The built-in software offers the possibility to program sequences where the instrument is switching between different settings. It usually takes ~10 s for $H_3O^+$ and ~60 s for $NO^+$ to re-stabilize after switching between the two measurement modes.

The ambient measurement data during the transition period (~1 min) is discarded. The voltages of ion source and drift chamber are changed in $H_3O^+$ (Us=150 V, Uso=80 V; Udrift=920 V,

Udx=46 V) and $NO^+$ (Us=40 V, Uso=100 V; Udrift=470 V, Udx=23.5 V) operating modes.

The pressures of drift chamber are held constant at 3.8 mbar in both modes during the campaigns.

The added and modified sentences on line 199-205 of page 9 are listed below:

**"Switching between $H_3O^+$ and $NO^+$ ion mode are provided by the PTR-MS Manager (v3.5)**

**software developed by the Ionicon Anlytik (Table S1). The pressures of drift chamber are**

**held constant at 3.8 mbar in both modes during the campaigns (Figure S10(a)). It usually**

**takes <10 s for $H_3O^+$ ions and ~60 s for $NO^+$ ions to re-stabilize after automatically**

**switching between the two measurement modes (Figure S10(b)). The ambient**

**measurement data during the transition period (~1 min) was discarded."**

Table S1. The settings of the voltages of ion source voltages (Us, Uso), drift tube (Udrift, Udx)

and pressure of drift tube (pDrift) during automatical switching between $NO^+$ mode and $H_3O^+$

mode, respectively.

| Setting | $NO^+$ mode | $H_3O^+$ mode |
|---|---|---|
| Us | 40 V | 150 V |
| Uso | 100 V | 80 V |
| Udrift | 470 V | 920 V |
| Udx | 23.5 V | 46 V |
| pDrift | 3.8 mbar | 3.8 mbar |

[Figure]

Figure S10. An example of the voltages of ion source voltages (Us, Uso), drift tube (Udrift, Udx) and pressure of drift tube (a), and the signal changes of primary ions (b) during automatical switching between NO$^+$ mode and H$_3$O$^+$ mode, respectively.

(3) This paper relies on predicted SOA production to compare the importance of alkanes versus aromatics and other compound classes. However, the SOA estimates rely on a number of assumptions (e.g., SOA yields) that are uncertain. There are other ways to compare the importance of alkanes to other compounds. There is very little discussion of absolute concentrations, which is one way to make the comparison. The authors could also compare something like OH reactivity.

Response: We included the comparison of average concentrations from higher alkanes (C8-C21 alkanes), monoaromatics (benzene, toluene, C8 aromatics, C9 aromatics and styrene), naphthalenes (naphthalene, methylnaphthalenes, dimethylnaphthalenes) and isoprenoids (isoprene and monoterpenes) in PRD and NCP in the revised supporting information. Compared to monoaromatics, higher alkanes have lower average concentrations. But due to the high SOA yields, higher alkanes also play an important role in SOA formation.

We added sentences on line 354-356 of page 15 in the revised manuscript as follows:

**"Compared to monoaromatics, higher alkanes are associated with lower concentrations (Figure S26). However, higher alkanes play an important role in SOA formation due to their high SOA yields (Figure S27)."**

[Figure]

Figure S26. The average concentrations from higher alkanes (C8-C21 alkanes), monoaromatics (benzene, toluene, C8 aromatics, C9 aromatics and styrene), naphthalenes (naphthalene, methylnaphthalenes, dimethylnaphthalenes) and isoprenoids (isoprene and monoterpenes) in PRD (a) and NCP (b), respectively.

[Figure]

Figure S27. Diurnal variations of SOA yields of *n*-C15 alkane, benzene, naphthalene and α-pinene in PRD (a) and NCP (b).

(4) The comparisons in Figure 8 (and associated discussion) do not seem to place alkanes and aromatics on equal footing. The alkane signal is essentially the sum of all C_x alkanes (i.e., C12 is the sum of dodecane, cyclododecane, and all branched isomers). This is analogous to integrating individual peaks in the GC along with the entire C12 "bin" of UCM (unresolved complex mixture) as shown by Zhao et al (2016a, b) and others. However, the authors only consider specific aromatics (e.g., BTEX) but not the aromatic UCM of larger substituted benzenes. A more complete comparison of alkanes to aromatics would include these species as well.

Response: Many thanks for your nice suggestion. Since we didn't have the measurement of aromatic UCM of larger substituted benzenes during these two campaigns, we only consider the specific aromatics (benzene, toluene, C8 aromatics, C9 aromatics and styrene) in this study. The aromatics used in this study are also the most considered aromatic hydrocarbon species in current SOA research. We agree that if the aromatic UCM of larger substituted benzenes are included, the contribution of aromatics to SOA may be even greater. However, in this study we want to emphasize the importance of higher alkanes to SOA formation, which is at least comparable to these common aromatic species, rather than figure out which VOCs class contribute most to SOA. The measurement of aromatic UCM of larger substituted benzenes and their contributions to SOA will also be considered in our future studies.

**Minor comments:**

(1) Lines 131-136 and Figure 2 - The figure would be easier to interpret if the relative humidity was also shown.

Response: The relative humidity is relative to the temperature. During the campaigns, especially in the NCP campaign, the relative humidity varies a lot due to the temperature changes. While, the sensitivities of higher alkanes are depended on the water vapour content in the sampling air. Previous studies indicate that water vapour in the gas phase can lead to the formation of hydrated hydronium ions $((H_2O)_n H_3O^+)$ in the drift chamber (Blake et al., 2009).

These hydrated hydronium ions can then themselves act as regent ions. Moreover, these hydrated hydronium ions can result in additional reactions (e.g. ligand switching) other than proton transfer in the drift chamber (Midey et al., 2000). Therefore, we use the absolute humidity to present the humidity dependence of higher alkanes in this paper.

(2) Line 192, in the comparison of GC-MS and $NO^+$ ionization for alkanes: "Similar temporal trends for these alkanes are observed from the two instruments." I can't really tell this from the figure. The diurnal trends suggest that the PTR-derived measurements have a deeper afternoon trough than the GC-MS, which seems to show a flatter concentration profile.

Response: We modified the figures to make the comparison clearer and easier to read. Since we don't have sufficient data and evidence, the deeper diurnal trends of higher alkanes in the afternoon from the PTR-derived measurements compared to GC-MS measurements are not discussed in this study.

[Figure]

Figure 5. Comparisons of times series and diurnal variations of alkanes measured by $NO^+$ PTR-ToF-MS and GC-MS/FID in PRD. **(a)** Time series of C8 alkanes measured by $NO^+$ PTR-ToF-MS, C8 *n*-alkane and four branched isomers measured by GC-MS/FID. **(b)** Diurnal variations of C8 alkanes. **(c-e)** Diurnal variations of C9-C11 alkanes with $NO^+$ PTR-ToF-MS and C9-C11 *n*-alkanes with GC-MS/FID.

(3) Line 193-194 - PTR data are a factor of 3-6 higher than the GC data. It looks like the GC data were only analyzed for species that are resolved by the GC (e.g., no UCM). How does the closure look if the entire chromatogram (or the entire m/z 57 signal) is binned and analyzed, similar to Zhao et al's (2016a, b) work?

Response: During the 2018 PRD campaign, the online GC-MS/FID system was operated in the selected ion monitoring (SIM) mode without scanning m/z 57 all the time. Therefore, we are not able to do the work as you suggested here.

(4) Why does Figure 5 only show PRD data for carbon numbers 8-11? The instrument measured up to C21.

Response: The online GC-MS/FID system we used in this campaign can only measure alkanes up to C11. So, we compared only C8-C11 alkanes, which were measured by both GC-MS/FID and $NO^+$ ToF-MS.

**Grammar:**

(1) Line 80 states that PTR "is response to large alkanes" - please edit. I think the authors mean that PTR responds to large alkanes.

Response: We modified "is response to large alkanes" to "is responsive to large alkanes".

(2) Line 160 - replace expect with except

Response: We replaced expect with except.

Reference:

Blake, R. S., Monks, P. S. and Ellis, A. M. Proton-Transfer Reaction Mass Spectrometry.
Chemical Reviews 2009, 109(3): 861-896.

Inomata, S., Tanimoto, H., and Yamada, H.: Mass Spectrometric Detection of Alkanes Using
$NO^+$ Chemical Ionization in Proton-transfer-reaction Plus Switchable Reagent Ion Mass
Spectrometry, Chemistry Letters, 43, 538-540, 10.1246/cl.131105, 2013.

Jordan, A., Haidacher, S., Hanel, G., Hartungen, E., Herbig, J., Maerk, L., Schottkowsky, R.,
Seehauser, H., Sulzer, P., and Maerk, T. D.: An online ultra-high sensitivity Proton-transfer-
reaction mass-spectrometer combined with switchable reagent ion capability (PTR+SRI-MS),
International Journal of Mass Spectrometry, 286, 32-38, 10.1016/j.ijms.2009.06.006, 2009.

Midey, A. J., Arnold, S. T. and Viggiano, A. A.. Reactions of $H_3O^+(H_2O)_n$ with Formaldehyde
and Acetaldehyde. The Journal of Physical Chemistry A, 2000, 104(12): 2706-2709.

---

## Author Comment (AC3) · 20 Aug 2020

**Response to anonymous referee #3**

In their manuscript, "Measurements of higher alkanes using $NO^+$ PTR-ToF-MS: significant contributions of higher alkanes to secondary organic aerosols in China," the authors tackle an important but often overlooked issue: the contribution of high-molecularweight gas-phase alkanes to SOA formation. A relatively large body of work over the last decade has provide a fairly thorough examination of the composition of vehicle emissions and the importance of branched (and cyclic) alkanes; this manuscript uses this work as a starting place to explore their contribution to SOA in China. The work is made possible in large part by the recent development of the $NO^+$ PTR. Overall, I think this work addresses interesting questions, and makes real efforts to tackle the complexity of calibrating a new instrument for compound classes that may contain hundreds of isomers. However, there are a few scientific weak spots that need to be addressed before this work is ready for publication. These larger comments include some needs for clarification, and some more fundamental issues with how OH exposure and SOA potential are being calculated. I commend the authors for taking a step into some unknown territory, but some additional discussion (and possibly work) needs to be included.

Response: Thank you very much for your valuable comments and suggestions on our manuscript. These comments are all helpful for improving our article. All the authors have seriously discussed about all these comments. According to your comments, we have tried best to modify our manuscript to meet with the requirements for the publication in this journal. The details about the calculation of OH exposure was added in the supporting information of the revised manuscript. Two methods to estimate SOA formation from different VOCs classes were all presented in the manuscript and the supporting information. Point-by-point responses to the comments are listed below.

**General Comments:**

(1) There are some typo and English issues throughout. It is not sufficient to seriously detract from the science, but it is to the degree that the authors should have a native English speaker review and edit this manuscript.

Response: Thanks for your nice suggestion.

(2) Considering the focus of this work is on large alkanes, and the semi-volatile nature of these components, particularly at the lower temperatures of some of these measurements, this manuscript really does not provide enough detail on the sample line and inlet. How long is this sample line? Is it heated all the way to the sample inlet, or just in the room? If it is not heated all the way to the inlet, I would have some misgivings about alkanes about C14 or so, there could be substantial losses or time lags for larger alkanes. Is there any evidence (observations or models) that suggest line losses and delays can be ignored? If the authors want to focus on gas-phase measurements of S/IVOC alkanes, more detail needs to be provided for the sampling.

Response: We included more information about the sampling in the revised manuscript. The schematic drawing of the inlet system for the measurement during campaigns is shown in the attached figure (Fig S2, Wu et al., 2020). The sampling line is ~8 m and ~3 m during PRD

campaign and NCP campaign respectively. Ambient air was continuously introduced into ToF-

MS through a Teflon tubing (1/4") with an external pump at 5.0 L/min. The calculated residence time for the inlet was ~3 s for PRD campaign and ~1 s for NCP campaign, respectively. The sampling line was heated all the way to the sampling inlet by an insulating tube with a self-controlled heater wire (40 ℃) wrapping outside. We conducted tubing loss experiments in the laboratory by introducing standards (2 ppb/1 ppb/0.5 ppb) of higher alkanes (*n*-C8-C15), monoaromatics (benzene, toluene, o-xylene, 1,2,4-trimethylbenzene), isoprenoids (isoprene, α-pinene) and naphthalene into PTR-ToF-MS through a 8 m Teflon tubing (1/4") at room temperature with an external pump at 5.0 L/min (Figure S11). The tubing loss of these standard compounds is found to be <5% except *n*-C15 of ~8% and naphthalene of ~10%. Given the fact that the sampling line was heated at 40 ℃ all the way to the inlet during the campaigns, we think the tubing loss would be ignored.

[Figure]

Figure S2. Schematic drawing of the inlet system for PTR-ToF-MS during the campaign. (Wu et al., 2020)

[Figure]

[Figure]

Figure S11. The tubing loss experiments of higher alkanes (*n*-C8-C15), 1,2,4-trimethylbenzene,α-pinene and naphthalene at room temperature using PTR-ToF-MS with an external pump at 5.0 L/min.

The sentences on line 205-216 of page 9-10 are modified as follows:

**"Ambient air was continuously introduced into PTR-ToF-MS through a Teflon tubing (1/4") with an external pump at 5.0 L/min, with tubing length of ~8 m and ~3 m during the PRD and the NCP campaign, respectively. The inlet tubing was heated all the way to the sampling inlet to avoid water vapour condensation by an insulating tube with a self-controlled heater wire (40 ℃) wrapping outside. The calculated residence time for the inlet was ~3 s for PRD campaign and ~1 s for NCP campaign, respectively. The tubing loss experiments were conducted in the laboratory by introducing standards of higher**

**alkanes (*n*-C8-C15), monoaromatics (benzene, toluene, o-xylene, 1,2,4-trimethylbenzene),**

**isoprenoids (isoprene, α-pinene) and naphthalene into PTR-ToF-MS through a 8 m**

**Teflon tubing (1/4") at room temperature with an external pump at 5.0 L/min (Figure**

**S11). The tubing loss of these compounds is found to be <5% except *n*-C15 (~8%) and**

**naphthalene (~10%)."**

3) In the discussion around "sensitivity", is that the response per mass of the m-1 ion, or the sum of all ions? If the former, it should be discussed in some of the relationships shown, as discussed in comments below. If the latter, how are those ions summed and attributed in the complex real-world data? Understanding of this terminology is critical for interpreting some of the figures.

Response: The "sensitivity" is that the response per mass of the m-1 ion with a unit of ncps/ppb in this study. We measured the fragmentation patterns of selected higher alkanes with $NO^+$

PTR-ToF-MS by introducing commercially acquired pure chemicals (Figure S8). We found that higher alkanes are mainly reacted through hydride abstraction by $NO^+$ forming mass (m-

1) ions (m is the molecular mass). The fractions of m-1 ions account for more than 60% of total ion signals for >C12 *n*-alkanes (Figure 4 (a)). Thus, we use the response per mass of the m-1

ion for each higher alkane when performing sensitivity experiments using a gas standard with a series of *n*-alkanes. We added a sentence on line 147-150 of page 7 in the revised manuscript to explain the "sensitivity" where this word was first mentioned.

**"Sensitivities of *n*-alkanes (C8-C15) standards were obtained during the campaign**

**(Figure S6), which is defined as the normalized signal of hydride abstraction ions for each**

**higher alkane at 1 ppbv with a unit of ncps/ppb."**

[Figure]

Figure S8. Mass spectra of the distributions of product ions from *n*-Dodecane **(a)**, *n*-
Pentadecane **(b)** and *n*-Eicosane **(c)** with NO$^+$ PTR-ToF-MS. The signals of masses shown in
the graph are the results after subtracting the isotopic signals during the high resolution peak
fitting of the mass spectra.

[Figure]

Figure 4. (a) The fractions of product ions (m-1) from hydride abstraction of C8-C20 *n*-alkanes in NO⁺ PTR-ToF-MS. (b) Scatterplot of sensitivities under dry conditions versus the fractions of hydride abstraction ions for C8-C15 *n*-alkanes.

4) More detail is necessary on how they calculate OH exposure. I'm not familiar with the isoprene chemistry method - please include a description. And for the ratio of aromatics method, don't you need to start by assuming some ratio at the time of emissions? What is assumed here?

Why does the OH exposure diurnal (Figure S9) look very different for the biogenics and the anthropogenics? Notably, in the NCP campaign this is also a large missing daytime source, could this be related to a "mistiming" of SOA caused by a bias in the OH exposure calculation?

Or what might be the missing source?

Response: We included more detail about the calculations of OH exposure in the revised supporting information. **The observed ratios between m+p-xylene and ethylbenzene were**

**used to estimate the OH exposure ($[OH] \times \Delta t$) by Roberts et al. (1984):**

$$[OH] \times \Delta t = \frac{1}{k_{\text{m+p-xylene}} - k_{ethylbenzene}} \times [ln(\frac{m + p - xylene}{ethylbenzene})_{t=0} - ln(\frac{m + p - xylene}{ethylbenzene})_{t}]$$

**Where the initial emission ratios of m+p-xylene/ethylbenzene were estimated according to**

**the correlation of m+p-xylene with ethylbenzene during campaigns. The ratio of 4 and**

**1.5 were used in the PRD campaign and the NCP campaign, respectively (Figure S29).**

[Figure]

Figure S29. Correlation of m+p-xylene with ethylbenzene in PRD (a) and NCP (b). The dashed lines in both graphs indicate the estimated initial mission ratio of m+p-xylene/ethylbenzene.

Isoprene are mainly photo-oxidized through the reactions with OH radical in the atmosphere and its primary first-generation reaction products are formaldehyde, MVK and MACR (Apel et al., 2002) . The reaction processes of isoprene oxidized by OH radical are mainly as follows:

$$Isoprene + OH \rightarrow 0.63HCHO + 0.32MVK + 0.23MACR \quad k_1=1.0\times10^{-10}\,cm^{-3}\,s^{-1} \ (Eq. \ S1)$$

$$MVK + OH \rightarrow Products \quad k_2=1.9\times10^{-11}\,cm^{-3}\,s^{-1} \ (Eq. \ S2)$$

$$MACR + OH \rightarrow Products \quad k_3=3.3\times10^{-11}\,cm^{-3}\,s^{-1} \ (Eq. \ S3)$$

where $k_1$, $k_2$, $k_3$ are the rate constants of the reactions. According to above reactions, the relationship between MVK/Isoprene, MACR/Isoprene, (MVK+MACR)/Isoprene and the reaction time $\Delta t$ can be decribed as follows (Apel et al., 2002):

$$\frac{MVK}{Isoprene} = \frac{0.32k_1}{k_2-k_1}\left(1 - \exp((k_1 - k_2)\,[OH]\Delta t)\right) \qquad (Eq. \ S4)$$

$$\frac{MACR}{Isoprene} = \frac{0.23k_1}{k_3-k_1}\left(1 - \exp((k_1 - k_3)\,[OH]\Delta t)\right) \qquad (Eq. \ S5)$$

$$\frac{MVK+MACR}{Isoprene} = \frac{0.32k_1}{k_2-k_1}\left(1 - \exp((k_1 - k_2)\,[OH]\Delta t)\right) + \frac{0.23k_1}{k_3-k_1}\left(1 - \exp((k_1 - k_3)\,[OH]\Delta t)\right) \quad (Eq. \ S6)$$

where $\frac{MVK+MACR}{Isoprene}$ can be derived from the measurements by PTR-ToF-MS. Then the OH exposure ($[OH]\Delta t$) of isoprenoids can be obtaind from Eq. S6.

The different diurnal patterns for the biogenics and the anthropogenics are mainly due to their different emission patterns. Much more fresh emissions for biogenic compounds during the daytime than nighttime, leading to the low OH exposure during the daytime for these compounds. The photochemical age of the air mass after mixing is the nonlinear addition of the photochemical age before mixing. Generally, the photochemical age of the mixed air mass is closer to that of the fresh air mass (Parrish et al., 1992). The large missing daytime source in the NCP might be some other SOA precursors or formation pathways (e.g. aqueous reactions) (Kuang et al., 2020)

The math on Eq. 1 makes sense, but I'm not sure the assumptions do. This assumes that the air behaves like a plug flow reactor from some emissions source, and then just oxidizes until the measurement site. Wouldn't local emissions (that perhaps have a different oxidation history) change things dramatically? Given that benzene and TMB aren't present in diesel or motor oil (which is the source of higher alkanes), this might be an issue. Though they are both from vehicle emissions, so maybe not. This might all be more compelling with more detail on how OH exposure is calculated.

Response: When calculating OH exposure for anthropogenic compounds, we use the ratio between m+p-xylene and ethylbenzene. Currently, given the fact that we don't have sufficient data to evaluate the specific contributions from different emission sources for all these anthropogenic compounds, we assume that m+p-xylene, ethylbenzene and higher alkanes have the same dominant emission sources during campaigns. In the near future, we will conduct VOCs measurements from typical emission sources in China.

Specific comments:

Line 135. The relationship between humidity and sensitivity seems fairly robust, so correctable, but I'm not sure a would classify a 50% drop in sensitivity as a "slight" decrease. If the "sensitivity" is to m-1, is the drop due to a change in fragmentation, or a drop in all ions? As it also due to a shift of $NO^+$ to $H_3O^+$ as water content increases, or does increasing humidity not increase $H_3O^+$?

Response: We modified the description about the relationship between humidity and sensitivity. We did several times humidity experiments in the laboratory by diluting higher alkanes standard into humidified air (relative humidity of 0-95% at 25℃) to reach approximately 1 ppb mixing ratio. The Figure 3 (b, c) summarize all the experiments data at different times to derive the relationships for C12 and C15 alkanes. The "sensitivity" is to m-1 ion. As shown in the attached Figure (a), the normalized signals of all product ions and the fragment ions are decreasing with the increase of humidity. These decreasing patterns are probably due to the decreasing reagent ions ($NO^+$ and $O_2^+$) as the humidity increases. Thus, the humidity correction should be applied for the quantitation of higher alkanes using $NO^+$ PTR-ToF-MS.

We modified and added sentences on line 163-168 of page 7-8 in the revised manuscript as follows:

**"As shown in Figure 3(b, c) and Figure S7 (a), the normalized signal of all product ions**

**(m-1) and the fragment ions of *n*-alkanes (C8-C15) standards are decreasing with the**

**increase of humidity. These decreasing patterns are probably due to the decreasing**

**primary reagent ions ($NO^+$ and $O_2^+$) as the humidity increases Figure S7(b). Thus, the**

**humidity correction should be applied for the quantitation of higher alkanes using $NO^+$**

**PTR-ToF-MS."**

[Figure]

Figure 3. (a) Calibrations of *n*-Dodecane and *n*-Pentadecane under dry conditions; (b) Humidity dependence of *n*-Dodecane. (c) Humidity dependence of *n*-Pentadecane.

[Figure]

Figure S7. Humidity dependence of all product ions and the fragment ions for *n*-alkanes (C8-C15) (a), and primary ions (NO$^+$, O$_2^+$, H$_3$O$^+$) (b).

Line 144. Is this slope between fraction of m-1 and sensitivity just due to the decrease m-1? In other words, is the total ion count produced the same, and just the fragmentation different, or is the total ion count lower for lower alkanes?

Response: We used a mixed standards of *n*-alkanes (C8-C15) to do the sensitivity experiments with NO$^+$ PTR-ToF-MS. Since almost all the higher alkanes produce the fragmentation ions of m/z 57, m/z 71, m/z 85, and m/z 99, we are not able to calculate the total ion count for each higher alkane with our experiments in this study. But the reason for this slope might be explored in the future when we have sufficient experimental conditions.

Line 144-145. Are calibrations of C16-C21 not available directly just because it is hard to get them into the gas phase, or was there some other reason?

Response: The commercial mixed standards of *n*-alkanes (C8-C15) we used in this study are all at 100 ppb except *n*-C15 at 30 ppb, because the lower vapour pressure of *n*-C15 make it difficult to obtain higher concentration of 100 ppb in the same cylinder. The vapour pressures of $n$-C16-$n$-C21 are even lower than that of $n$-C15. It is not possible to prepare $n$-C16-$n$-

C21standards with concentrations of $>$ 30 ppb into the mixed standards of $n$-alkanes (C8-C15)

in the same cylinder. Moreover, the very low vapour pressure compounds are not stable in the cylinder.

Line 160. Should be "except" instead of "expect Line 162. How was this switching achieved?

Was it just a change in voltages? Are there residual effects observed, and/or do some transitional data need to be removed?

Response: We replaced the "expect" with "except" in the revised manuscript. More details about the automatic switches between $H_3O^+$ chemistry and $NO^+$ chemistry are also included on line 199-205 of page 9 in the revised paper.

**"Switching between $H_3O^+$ and $NO^+$ ion mode are provided by the PTR-MS Manager (v3.5)**

**software developed by the Ionicon Anlytik (Table S1). The pressures of drift chamber are**

**held constant at 3.8 mbar in both modes during the campaigns (Figure S10(a)). It usually**

**takes <10 s for $H_3O^+$ ions and ~60 s for $NO^+$ ions to re-stabilize after automatically**

**switching between the two measurement modes (Figure S10(b)). The ambient**

**measurement data during the transition period (~1 min) was discarded."**

Table S1. The settings of the voltages of ion source voltages (Us, Uso), drift tube (Udrift, Udx)

and pressure of drift tube (pDrift) during automatical switching between $NO^+$ mode and $H_3O^+$

mode, respectively.

| Setting | $NO^+$ mode | $H_3O^+$ mode |
|---|---|---|
| Us | 40 V | 150 V |
| Uso | 100 V | 80 V |
| Udrift | 470 V | 920 V |
| Udx | 23.5 V | 46 V |
| pDrift | 3.8 mbar | 3.8 mbar |

[Figure]

Figure S10. An example of the voltages of ion source voltages (Us, Uso), drift tube (Udrift, Udx) and pressure of drift tube (a), and the signal changes of primary ions (b) during automatical switching between $NO^+$ mode and $H_3O^+$ mode, respectively.

Line 167. I'm not quite sure what the "insulated tube" is - do they mean heater rope or something?

Response: We revised the sentence. "**The inlet tubing was heated all the way to the sampling inlet to avoid water vapour condensation by an insulating tube with a self-controlled heater wire (40 ℃) wrapping outside.**"

Line 185. Do the authors mean there are limitations to its application? Or just that it hasn't been applied very often?

Response: We revised the sentence. "**Although $NO^+$ chemistry has been shown to be valuable in measuring many organic species, the applications in real atmosphere of different environments are still rare**".

Line 187-190. The comparison between H3O+ and NO+ and GC-MS/FID is very encouraging.
There are some significantly non-unity slopes though (0.47 for benzene) -do the authors have
an explanation for this?

Response: Yes, the benzene of $H_3O^+$ measurement is not well consistent with that of $NO^+$
measurement in the formal period of PRD campaigns. We don't know the exact reason for this.
Considering the good consistency of benzene between $NO^+$ measurement and GC-MS
measurement, we used the benzene from $NO^+$ measurement in this study.

Line 276. A Table of SOA yields used would be helpful for the SI

Response: We included a table of SOA yields in the revised supporting information (Table S3).

Table S3. The calculated average SOA yields of higher alkanes in PRD and NCP.

| Compounds | Formula | Average SOA | Average SOA |
|-----------|---------|-------------|-------------|
| Octane | $C_8H_{18}$ | 0.003±0.002 | 0.006±0.004 |
| Nonane | $C_9H_{20}$ | 0.010±0.005 | 0.017±0.010 |
| Decane | $C_{10}H_{22}$ | 0.026±0.012 | 0.040±0.021 |
| Undecane | $C_{11}H_{24}$ | 0.058±0.020 | 0.080±0.036 |
| Dodecane | $C_{12}H_{26}$ | 0.106±0.032 | 0.142±0.059 |
| Tridecane | $C_{13}H_{28}$ | 0.249±0.061 | 0.305±0.103 |
| Tetradecane | $C_{14}H_{30}$ | 0.329±0.070 | 0.388±0.118 |
| Pentadecane | $C_{15}H_{32}$ | 0.386±0.081 | 0.450±0.135 |
| Hexadecane | $C_{16}H_{34}$ | 0.428±0.086 | 0.492±0.141 |
| Heptadecane | $C_{17}H_{36}$ | 0.488±0.096 | 0.556±0.156 |
| Octadecane | $C_{18}H_{38}$ | 0.664±0.079 | 0.704±0.139 |
| Nonadecane | $C_{19}H_{40}$ | 0.773±0.056 | 0.792±0.105 |
| Eicosane | $C_{20}H_{42}$ | 0.860±0.025 | 0.863±0.054 |
| Heneicosane | $C_{21}H_{44}$ | 0.877±0.025 | 0.870±0.046 |

Line 301-304. Are there emissions sources for only alkanes? I would guess not, in which case
these alkanes must be getting emitted alongside cyclic saturated hydrocarbons (e.g.,
cycloalkanes), which actually contribute similar or more to petroleum fuels (e.g., Gentner et
al., 2012, Table S5). These compounds are expected to have broadly similar (or slightly higher)
SOA yields. This would have you substantially overestimating SOA in NCP at night - the
authors should comment on possible explanations or biases.

Response: In this study, the higher alkanes represent only the total concentration of linear and
branched isomers, without considering the cyclic alkanes. Yes, some studies have indicated
that cyclic alkanes also contribute importantly to SOA formation (Tkacik et al., 2012; Gentner
et al., 2012). In the near future, we will also try the possibility to measure these cyclic alkanes.
Including these cyclic alkanes would enhance the estimation of SOA production shown in this
study.

Figure 5. I'm not quite sure what data is being shown here. Gentner provides diesel fuel data
in table S5 of that publication (this is used to approximate exhaust in that manuscript). That
data doesn't agree with what is shown here though - for C8 it is 37%, and all the values are
near or above 21%. The Isaacman paper is actually just a single fuel sample that is included in
the Gentner data set, so could probable be excluded. That paper also provides gasoline data,
which is not shown. In Figure 9 of the Chan paper, the branched-to-normal ratios are given for
vehicle exhaust, which convert to between 13 and 41% normal for C24 through C21, which
are fairly different than those shown. Similarly, in Figure 5, the branched-to-normal ratios for
LA are shown at multiple oxidation times, and reach levels of 20 to 50% for those same carbon
numbers - this is simlar to the values shown, but I think not quite right. I think the numbers for
Bakersfield (based on Figure 4 from that paper) should be much lower, 5-10%, for all carbon
numbers. Values should also be included for direct vehicle emissions from Worton et al., 2014,
dx.doi.org/10.1021/es405375j. Basically, some explanation for how these numbers were all
compiled is necessary (even if it's just in the SI), because they don't look quite right to me.

Response: Linear alkanes and branched alkanes follow the same molecular formula: $C_nH_{2n+2}$.
When using $NO^+$ ToF-MS, we measure the total concentrations of linear alkanes and branched
alkanes with the same formulas since this technique doesn't differentiate isomers. We collected
data from literatures which are derived from GC-based techniques to calculate the mass
fractions of $n$-alkanes in higher alkanes with the same formula from various ambient and
emission studies, with the purpose of emphasizing the importance of contribution of branched isomers to higher alkanes concentrations at each carbon number. The fraction of *n*-alkane at each carbon number is calculated as follows:

$$Fraction_{linear} = \frac{Concentration_{linear}}{Concentration_{linear} + Concentration_{branched}} \times 100\%$$

As shown in the figure, *n*-alkanes contribute ~5-60% to higher alkanes concentrations from C8 to C25, indicating that branched isomers contribute up to ~40-95%. This indicates the measurement of branched isomers is also important. We checked the data collected from references and modified the figure (Figure 6) as suggested. A table of data used in this figure was also included in the revised supporting information (Table S2). The data from Worton et al., 2014 was not included because those high alkanes data are particle phase. In this study, we only focus on the gas phase alkanes and their importance to SOA.

[Figure]

Figure 6. Fractions of *n*-alkanes in higher alkanes with same formulas derived from this study, ambient air in Los Angeles, Bakersfield and in vehicle exhausts.

 Table S2. Fractions of *n*-alkanes in higher alkanes with same formulas derived from this study, ambient air in Los Angeles, Bakersfield and in vehicle exhausts.

| Carbon Number | Fraction of *n*-alkanes in higher alkanes with same formulas | | | | |
|---|---|---|---|---|---|
| | PRD[a] | Los Angeles[b] | Bakersfield[b] | Diese exhaust[c] | Liquid gasoline exhaust[c] |
| 8 | 10.82 | / | / | 37.04 | 5.39 |
| 9 | 21.48 | / | / | 51.22 | 7.71 |
| 10 | 17.56 | / | / | 23.81 | 8.81 |
| 11 | 17.81 | / | / | 20.91 | 10.88 |
| 12 | / | / | / | 22.54 | 29.82 |
| 13 | / | / | / | 21.98 | / |
| 14 | / | / | / | 19.84 | 5.41 |
| 15 | / | / | / | 22.86 | 40 |
| 16 | / | / | / | 25.44 | / |
| 17 | / | / | / | 32.16 | / |
| 18 | / | / | / | 28.57 | / |
| 19 | / | / | / | 20.83 | / |
| 20 | / | 34.78 | 24.85 | 20.87 | / |
| 21 | / | 53.16 | 12.47 | 24.82 | / |
| 22 | / | 42.85 | 9.11 | 25.51 | / |
| 23 | / | 40.24 | 8.14 | 21.05 | / |
| 24 | / | 21.85 | 6.92 | 24.44 | / |
| 25 | / | 27.17 | 27.87 | 60 | / |

[a]:This work; [b]: Chan et al., 2013; [c]: Gentner et al., 2012

Reference:

Apel, E. C., Riemer, D. D., Hills, A., Baugh, W., Orlando, J., Faloona, I., Tan, D., Brune, W.,
Lamb, B., Westberg, H., Carroll, M. A., Thornberry, T., and Geron, C. D.: Measurement and
interpretation of isoprene fluxes and isoprene, methacrolein, and methyl vinyl ketone mixing
ratios at the PROPHET site during the 1998 Intensive, Journal of Geophysical Research:
Atmospheres, 107, ACH 7-1-ACH 7-15, 10.1029/2000JD000225, 2002.

Chan, A. W. H., Isaacman, G., Wilson, K. R., Worton, D. R., Ruehl, C. R., Nah, T., Gentner,
D. R., Dallmann, T. R., Kirchstetter, T. W., Harley, R. A., Gilman, J. B., Kuster, W. C.,
deGouw, J. A., Offenberg, J. H., Kleindienst, T. E., Lin, Y. H., Rubitschun, C. L., Surratt, J.
D., Hayes, P. L., Jimenez, J. L., and Goldstein, A. H.: Detailed chemical characterization of
unresolved complex mixtures in atmospheric organics: Insights into emission sources,
atmospheric processing, and secondary organic aerosol formation, Journal of Geophysical
Research-Atmospheres, 118, 6783-6796, 10.1002/jgrd.50533, 2013.

Gentner, D. R., Isaacman, G., Worton, D. R., Chan, A. W. H., Dallmann, T. R., Davis, L., Liu,
S., Day, D.A., Russell, L. M., Wilson, K. R., Weber, R., Guha, A., Harley, R. A., and Goldstein,
A. H.: Elucidating secondary organic aerosol from diesel and gasoline vehicles through
detailed characterization of organic carbon emissions, Proceedings of the National Academy
of Sciences of the United States of America, 109, 18318-18323, 10.1073/pnas.1212272109,
2012.

Kuang, Y., He, Y., Xu, W., Yuan, B., Zhang, G., Ma, Z., Wu, C., Wang, C., Wang, S., Zhang,
S., Tao, J., Ma, N., Su, H., Cheng, Y., Shao, M., and Sun, Y.: Photochemical Aqueous-Phase
Reactions Induce Rapid Daytime Formation of Oxygenated Organic Aerosol on the North
China Plain, Environmental Science & Technology, 10.1021/acs.est.9b06836, 2020.

Roberts, J. M., Fehsenfeld, F. C., Liu, S. C., Bollinger, M. J., Hahn, C., Albritton, D. L., and
Sievers, R. E.: Measurements of aromatic hydrocarbon ratios and NOx concentrations in the
rural troposphere: Observation of air mass photochemical aging and NOx removal,
Atmospheric Environment (1967), 18, 2421-2432, https://doi.org/10.1016/0004-
6981(84)90012-X, 1984.

Parrish, D. D., C. J. Hahn, E. J. Williams, R. B. Norton, F. C. Fehsenfeld, H. B. Singh, J. D.
Shetter, B. W. Gandrud and B. A. Ridley. Indications of photochemical histories of Pacific air masses from measurements of atmospheric trace species at Point Arena, California. Journal of

Geophysical Research: Atmospheres, 1992, 97(D14): 15883-15901.

Roberts, J., Marchewka, M., Bertman, S., Goldan, P., Kuster, W., de Gouw, J., warneke, C.,

Williams, E., Lerner, B., Murphy, P., Apel, E., and Fehsenfeld, F.: Analysis of the isoprene chemistry observed during the New England Air Quality Study (NEAQS) 2002 Intensive

Experiment, Journal of Geophysical Research-Atmospheres, 111, D23S12,

10.1029/2006JD007570, 2006.

Tkacik, D. S., Presto, A. A., Donahue, N. M., and Robinson, A. L.: Secondary Organic Aerosol

Formation from Intermediate-Volatility Organic Compounds: Cyclic, Linear, and Branched

Alkanes, Environmental Science & Technology, 46, 8773-8781, 10.1021/es301112c, 2012.

Wu, C., Wang, C., Wang, S., Wang, W., Yuan, B., Qi, J., Wang, B., Wang, H., Wang, C., Song,

W., Wang, X., Hu, W., Lou, S., Ye, C., Peng, Y., Wang, Z., Huangfu, Y., Xie, Y., Zhu, M.,

Zheng, J., Wang, X., Jiang, B., Zhang, Z., and Shao, M.: Measurement Report: important contributions of oxygenated compounds to emissions and chemistry of VOCs in urban air,

Atmos. Chem. Phys. Discuss., 2020, 1-37, 10.5194/acp-2020-152, 2020.

---

## Author Comment (AC4) · 20 Aug 2020

**Response to anonymous referee #4**

**General comments:**

This work is a nice, extensive manuscript using an $NO^+$ CIMS (see below) to measure ambient alkanes in the PRD of China and evaluate their importance to SOA chemistry. The experimental methods are high quality and very well-documented. The importance of this manuscript is high, and I find it to be novel and useful.

It is, however, difficult to understand in places and suffers from some lack of precision in language. I recommend publication after these relatively minor, but common issues are fixed.

Response: Thank you very much for your valuable comments and suggestions on our manuscript. These comments are all helpful for improving our article. All the authors have seriously discussed about all these comments. According to your comments, we have tried best to modify our manuscript to meet with the requirements for the publication in this journal. Point-by-point responses to the comments are listed below.

**Specific comments:**

(1) Instrument name: The title of the paper and description of the instrument is not correct. PTR is a widely used term that refers to the transfer of a proton from $H_3O^+$ under controlled conditions. $NO^+$ ionization uses different ionization mechanisms, particularly charge transfer. I understand the instrument is a commercial PTR-MS, but the authors are now using a different reagent ion and the authors should change the name to something like $NO^+$ ToF-MS or $NO^+$ CIMS. Analytical chemistry acronyms are confusing enough as it is. Removing all meaning from them will make them unintelligible to everyone.

Response: We understand the reviewer's concern. We use the $NO^+$ PTR-TOF-MS as the instrument name to reflect the fact that $NO^+$ chemistry has been mainly adopted from PTR-MS instruments as a switchable reagent ion technique (Jordan et al. 2009), which is also the case for this work. The first attempt to measure higher alkanes using $NO^+$ chemistry (Inomata et al., 2013) referred this method as $NO^+$ chemical ionization using proton-transfer-reaction mass spectrometry, which seems to reflect the reality of both the ion chemistry and also the instrumentation used for the method. As the result, we changed the title in the revised manuscript: "**Measurements of higher alkanes using $NO^+$ chemical ionization in PTR-ToF-MS: important contributions of higher alkanes to secondary organic aerosols in China**".

In the main text (line 98-99) of the revised manuscript, we also reflect this information:

**"In this study, we utilize NO$^+$ chemical ionization in PTR-ToF-MS (here referred as NO$^+$ PTR-TOF-MS) to measure higher alkanes at two different sites in China."**

(2) SOA Yields: There would appear to be a large amount of possible error in these measurements and calculations and it needs to be reflected when yields (SOA production) are displayed across the paper. Some yield calculations are presented with errors and some are not. Figure 7 and 8, in particular have issues with this.

Response: We included the errors when presenting the concentrations, SOA yields and SOA production. The errors displayed in this study refer to the standard deviations (1δ) over the averaging period during the campaigns.

(3) P2 L29: Is it novel? This method has previously been published (with GC comparisons) by some of the co-authors of this paper in Koss et al.

Response: We deleted the word "novel".

(4) P12 L280: Is this really a surprise? Why? Do those references use different yields or inputs than this work?

Response: We deleted the word "interesting and the sentence was modified to **"The distribution of contributions from alkanes with different carbon number to SOA formation shown here is in good agreement with the previous results referred from volatility calculation for precursors (de Gouw et al., 2011;Liggio et al., 2016)**."

(5) Figure 1: The mass-to-charge labels on the top of the graph are entirely illegible and the labels on the axes are too small to be conveniently legible. Please make larger. I understand that the authors used the default Tofware labels, but "IndivHRfitCurves" will not mean a lot to most readers. Please change the labels to something clearer in each graph.

Response: We modified this figure in the revised manuscript as follows.

[Figure]

Figure 1. High-resolution (HR) peak-fitting to the averaged mass spectra on a typical day (12 October 2018) for m/z 169 (a), m/z 211 (b) and m/z 281 (c), at which masses produced by dodecane ($C_{12}H_{25}^+$), pentadecane ($C_{15}H_{31}^+$) and eicosane ($C_{20}H_{41}^+$) produced in $NO^+$ PTR-ToF-MS.

(6) Figure 5 gives the impression that the authors suggest the dominant source of alkanes in the region is diesel vehicles. Is that correct? If not, I'm not sure I understand the point of putting those traces on the graph with ambient measurements.

Response: This figure was used to emphasize the importance of contribution of branched isomers to higher alkanes concentrations at each carbon number. Linear alkanes and branched alkanes follow the same molecular formula: $C_nH_{2n+2}$. When using $NO^+$ PTR-ToF-MS, we measure the total concentrations of linear alkanes and branched alkanes with the same formulas since this technique doesn't differentiate isomers. We collected data from literatures which are derived from GC-based techniques to calculate the mass fractions of *n*-alkanes in higher alkanes with the same formula from various ambient and emission studies. Because of quite limited studies published about the emission sources of higher alkanes, we only collected few data about the vehicle exhaust. The other emission sources of higher alkanes are still unknown and are needed to study in the future.

[Figure]

Figure 6. Fractions of *n*-alkanes in higher alkanes with same formulas derived from this study, ambient air in Los Angeles, Bakersfield and in vehicle exhausts.

(7) Figure 7a needs error bars on the calculated SOA yields. There would appear to be a large amount of possible error in these measurements and calculations and it needs to be reflected when yields are displayed in this work

Response: We included the error bars on the calculated SOA yields in Figure 8 in the revised manuscript as follows. In this study, SOA yield data for higher alkanes (Lim and Ziemann,

2009;Presto et al., 2010;Loza et al., 2014;Lamkaddam et al., 2017b) were summarized from reported values in the literature, with the consideration of the influence of organic aerosol concentration to SOA yields (Figure S20). The error bars in Figure 8(a) refer to standard deviations (1δ) over the averaging period of calculated SOA yields. The error bars in Figure

8(b) refer to standard deviations (1δ) over the averaging period of calculated SOA production during the campaigns.

[Figure]

Figure S20. The reported SOA yields as a function of OA concentrations for higher alkanes (C8-C21 alkanes) (a-k) under high-$NO_x$ condition from chamber studies.

[Figure]

Figure 8. (a) Measured concentrations by NO⁺ PTR-ToF-MS, calculated consumed concentrations and average SOA yields for C8-C21 alkanes in PRD and NCP. **The error bars represent standard deviations (1δ) over the averaging period of calculated SOA yields** (b) Calculated average SOA productions for C8-C21 alkanes in PRD and NCP. **The error bars represent standard deviations (1δ) over the averaging period of calculated SOA production**.

**Technical corrections:**

P4 L80: should be "is responsive" P4 L82: Not PTR-MS. See above. P4 L88: "in more detail." P5 L102: See above. Not PTR-MS P5 L110: "mass resolving power" instead of resolution P5 L111: "source" P15 L343: "While: : :" this sentence needs to be re-written for grammatical clarity

Many small language clarity issues throughout the manuscript.

Response: Thanks. We corrected all these technical issues and checked the language clarity issues throughout the manuscript.

Reference:

Inomata, S., Tanimoto, H., and Yamada, H.: Mass Spectrometric Detection of Alkanes Using
$NO^+$ Chemical Ionization in Proton-transfer-reaction Plus Switchable Reagent Ion Mass
Spectrometry, Chemistry Letters, 43, 538-540, 10.1246/cl.131105, 2013.

Jordan, A., Haidacher, S., Hanel, G., Hartungen, E., Herbig, J., Maerk, L., Schottkowsky, R.,
Seehauser, H., Sulzer, P., and Maerk, T. D.: An online ultra-high sensitivity Proton-transfer-
reaction mass-spectrometer combined with switchable reagent ion capability (PTR+SRI-MS),
International Journal of Mass Spectrometry, 286, 32-38, 10.1016/j.ijms.2009.06.006, 2009.

---

## Author Comment (AC5) · 20 Aug 2020

**Response to anonymous referee #5:**

Wang et al. report on measurements of higher alkanes made with a PTR-ToF-MS in an urban and rural location in China. They find relatively abundant quantities of higher alkanes and, based on estimates of OH concentrations and SOA mass yields, argue that they also contribute meaningfully to SOA formation in both regions. Measuring the chemical composition and complexity of VOCs in the atmosphere is an important step in studying the potential of those VOCs to form SOA and identify the sources of fine particulate matter. The study is well motivated and mostly well written (some detailed comment about technical communication can be found below). I am not overtly familiar with the online mass spectrometry measurements so the editor should rely on the other reviews to make a judgement about that. The data analysis and modeling sections require some additional detail to better communicate the inputs and assumptions. I generally favor publication of this work in ACP but after the authors have had a chance to respond to my comments.

Response: Thank you very much for your valuable comments and suggestions on our manuscript. These comments are all helpful for improving our article. All the authors have seriously discussed about all these comments. According to your comments, we have tried best to modify our manuscript to meet with the requirements for the publication in this journal. Point-by-point responses to the comments are listed below.

**Comments:**

1. Line 2: After reviewing the manuscript, I did not find that the word 'significant' was appropriate in the title. The authors find that alkanes are probably as important as aromatics and biogenic VOCs but the model currently still underestimates the total SOA production in both studied regions. The word 'significant' could be misconstrued to mean that alkanes explain the majority of the SOA in urban and rural china.

Response: We replaced the word "significant" with "important".

2. Lines 47-53: This introduction to SOA modeling is not well described and does accurately represent the historical approaches used to model SOA formation. For instance, precursor lumping has been used prior to the volatility basis set.

Response: We modified this part of introduction as follows. "**A volatility basis set (VBS) model was developed to advance SOA modeling by improving the modeling of further multigenerational oxidation processes and incorporating numerous, yet unidentified,**

**low-volatility precursors of SOA, which substantially improved the agreement between SOA simulations and observations.**"

3. Lines 58-61: It might be worthwhile to talk about the differences in the various SOA precursor classes between gasoline and diesel exhaust.

Response: We included the introduction about the differences in the various SOA precursor classes between gasoline and diesel exhaust on line 69-71 of page 4 in the revised manuscript.

**"Based on vehicle exhaust tests, higher alkanes were found to contribute ~37% to diesel exhaust-derived SOA and ~0.8% to gasoline exhaust-derived SOA, respectively (Gentner et al., 2012).**"

4. Lines 61-63, lines 314-316: A recent study by Akherati et al. (ACP, 2019) also modeled SOA formation from higher alkanes but did not find them to be as important when compared to aromatics, especially after accounting for the effects of vapor wall losses (see #12).

Response: The estimation of SOA formation from higher alkanes in this study is calculated based on the field measurements in the urban and rual regions of China, which is different from the model study in Akherati et al. (ACP, 2019). According to the calculated SOA yield biases for a variety of VOCs when considering the potential influence of vapour wall losses (Table S4), the literature reported SOA yields are low by factors of ~1.1-2.2 for the high $NO_x$ conditions (Zhang et al., 2014). This suggests the SOA calculations in our study might be correspondingly underestimated. However, we still can find the importance of higher alkanes to SOA formation in our study.

Table S4. Average biases in SOA yields due to vapour wall losses for various VOCs under high-$NO_x$ conditions from Zhang et al. 2014.

| VOC | $R_{wall}$ |
| --- | --- |
| Benzene | 1.25±0.1 |
| Toluene | 1.13±0.06 |
| m-xylene | 1.2±0.1 |
| Naphthalene | 1.2±0.1 |
| Isoprene | 2.2±0.5 |
| α-pinene | 1.3±0.1 |
| n-dodecane | 1.16±0.08 |
| 2-methylundecane | 1.4±0.2 |

5. Line 80: 'is response' should be 'responds'.

Response: We corrected it to "responsive".

6. Line105: grammatical mistake in 'makes possible of quantitative of alkenes'.

Response: We corrected this sentence.

7. Lines 147-148: grammatical mistake in 'group affect little on the degrees of fragmentation
for product ions'.

Response: We corrected this sentence.

8. Lines 193-195: I don't see why the PTR measurement should be higher than the GC.
Shouldn't the GC measurement be close to the PTR if the unresolved mixture is accounted for?

Response: In this study, the higher alkanes standards used by online GC-MS are just linear
alkanes (*n*-alkanes) without branched ones. Therefore, the quantification of branched alkanes
by GC-MS is not available in this study. While, the higher alkanes measured by PTR-ToF-MS
include linear and branched isomers for each carbon number.

9. Line 212: Should be 'This decreasing pattern'.

Response: We corrected this sentence.

10. Lines 239-242: It isn't clear to me why different pairs of species are used to estimate OH
concentrations for the anthropogenic and biogenic species separately. Can the equation for the
OH estimation be provided? Also, is there confounding in the OH estimation associated with
emissions being added to the air parcel while the selected pairs are oxidized?

Response: We included the details about the calculations of OH exposure in the revised
supporting information. During the PRD campaign, since biogenic species (i.e. isoprene and
monoterpenes) have different emission sources and patterns from the anthropogenic species,
the photochemical oxidation processes are quite different in these species. Therefore, we use
different methods to calculate the OH exposures for anthropogenic species and biogenic
species during PRD campaign. For the anthropogenic species, the observed ratio between m+p-
xylene and ethylbenzene was used to estimate the OH exposure ( $[OH] \times \Delta t$ ) by Roberts et al.
(1984) :

$$[OH] \times \Delta t = \frac{1}{k_{\text{m+p-xylene}} - k_{ethylbenzene}} \times [ln(\frac{\text{m+p-xylene}}{ethylbenzene})_{t=0} - ln(\frac{\text{m+p-xylene}}{ethylbenzene})_{t}]$$

Where the initial emission ratios of m+p-xylene/ethylbenzene were estimated according
to the correlation of m+p-xylene with ethylbenzene during campaigns. The ratio of 4 and 1.5
were used in the PRD campaign and the NCP campaign, respectively (Figure S29).

[Figure]

Figure S29. Correlation of m+p-xylene with ethylbenzene in PRD (a) and NCP (b). The dashed
lines in both graphs indicate the estimated initial mission ratio of m+p-xylene/ethylbenzene.

The OH exposure of biogenic emissions are estimeted based on isoprene chemistry for
PRD campaign. The calculation method can be found in Roberts et al. (2006). Isoprene are
mainly photo-oxidized through the reactions with OH radical in the atmosphere and its primary
first-generation reaction products are formaldehyde, MVK and MACR (Apel et al., 2002) . The
reaction processes of isoprene oxidized by OH radical are mainly as follows:

*Isoprene + OH → 0.63HCHO + 0.32MVK + 0.23MACR*    *$k_1=1.0\times10^{-10}\,cm^{-3}\,s^{-1}$ (Eq. S1)*

*MVK + OH → Products*    *$k_2=1.9\times10^{-11}\,cm^{-3}\,s^{-1}$ (Eq. S2)*

*MACR + OH → Products*    *$k_3=3.3\times10^{-11}cm^{-3}s^{-1}$ (Eq. S3)*

where $k_1$, $k_2$, $k_3$ are the rate constants of the reactions. According to above reactions, the
relationship between MVK/Isoprene, MACR/Isoprene, (MVK+MACR)/Isoprene and the
reaction time $\Delta t$ can be decribed as follows (Apel et al., 2002):

$$\frac{MVK}{Isoprene} = \frac{0.32k_1}{k_2-k_1}\left(1 - \exp((k_1 - k_2)\,[OH]\Delta t)\right) \qquad (Eq.S\ 4)$$

$$\frac{MACR}{Isoprene} = \frac{0.23k_1}{k_3-k_1}\left(1 - \exp((k_1 - k_3)\,[OH]\Delta t)\right) \qquad (Eq.\ S5)$$

$$\frac{MVK+MACR}{Isoprene} = \frac{0.32k_1}{k_2-k_1}\left(1 - \exp((k_1 - k_2)\,[OH]\Delta t)\right) + \frac{0.23k_1}{k_3-k_1}\left(1 - \exp((k_1 - k_3)\,[OH]\Delta t)\right) \quad (Eq.$$

*S6)*

where $\frac{MVK+MACR}{Isoprene}$ can be derived from the measurements by PTR-ToF-MS. Then the OH

exposure ($[OH]\Delta t$) of isoprenoids can be obtain from Eq. S6.

The OH exposure values in this study represent the average "photochemical age" that are calculated by some properly chosen VOCs ratios. Parrish et al. (2007) compared the calculated photochemical age of different NMHCs species with the transport time calculated by a back trajectory model, and concluded that the calculated photochemical age could better describe the average transport time of VOCs species by selecting suitable hydrocarbon species pairs.

11. Lines 242-246: If the isoprene and monoterpenes are of anthropogenic origin, shouldn't the biogenic VOCs be oxidized based on the OH determined from the anthropogenic VOCs?

Response: In this study, we think the isoprene and monoterpenes are of anthropogenic origin in NCP. Thus, we use the OH exposure estimated from the anthropogenic VOCs for isoprenoids in NCP campaign. While, because the isoprenoids in PRD campaign are of biogenic origin, we estimate the OH exposure of isoprenoids based on isoprene chemistry method as mentioned above.

12. Lines 253-onwards: While I commend the authors for relying on a lot of historical data to determine SOA parameterizations for the VOCs measured in this work, the toluene parameterizations do not use the more recent work of Zhang et al. (PNAS, 2014) that account for the influence of vapor wall losses in chambers. On a related note, were the SOA mass yields adjusted for vapor wall losses? Alternatively, some of the parameterizations can also be found in Eluri et al. (ACP, 2018). Finally, how was the NOx-dependence on SOA modeled? No NOx data at either of those sites was presented, nor any argument was made for the use of high NOx

SOA parameters.

Response: Thank you for the nice suggestion. We agree with that the vapour wall losses might influence the calculation of SOA formation by using SOA yields from previous chamber studies where the wall losses were rarely corrected. Cappa and Wilson (2012) developed a

Statistical Oxidation Model (SOM) to simulate the multigenerational chemistry and gas- particle partitioning of organic compounds. This SOM has been used to interpret chamber experiments to examine the influence of chamber-based vapour wall losses on yields of SOA

(Zhang et al., 2014). According to the calculated SOA yield biases for a variety of VOCs when considering the potential influence of vapour wall losses (Table 1 in this reference), the literature reported SOA yields are low by factors of ~1.1-2.2 for the high $NO_x$ conditions
(Zhang et al., 2014). This suggests the SOA calculations in our study might be correspondingly
underestimated. In the revised manuscript on line 370-374 of page 16, we added sentences to
mention this influence as follows:

**"The influence of chamber-based vapour wall losses on SOA yields was examined in**
**previous studies (Zhang et al., 2014) and the results show that the literature reported**
**SOA yields are low by factors of ~1.1-2.2 for the high $NO_x$ conditions (Table S4). This**
**suggests that the SOA estimations in this study might be correspondingly underestimated."**

Table S4. Average biases in SOA yields due to vapour wall losses for various VOCs under
high-$NO_x$ conditions from Zhang et al. 2014.

| VOC | $R_{wall}$ |
|---|---|
| Benzene | 1.25±0.1 |
| Toluene | 1.13±0.06 |
| m-xylene | 1.2±0.1 |
| Naphthalene | 1.2±0.1 |
| Isoprene | 2.2±0.5 |
| α-pinene | 1.3±0.1 |
| n-dodecane | 1.16±0.08 |
| 2-methylundecane | 1.4±0.2 |

We included the $NO_x$ data in the revised supporting information (Figure S22). The
average concentrations of $NO_x$ are 42.6±33.7 ppb in PRD and 81.7±57.0 ppb in NCP,
respectively. Under such high $NO_x$ levels, the $RO_2$ mainly reacts with NO (Bahreini et al.,
2009). Thus, we used the high SOA parameters in this study.

We added sentences on line 321-324 of page 14 in the revised manuscript as follows:

**"SOA yields under high $NO_x$ conditions are used in this study, as relatively high $NO_x$**
**concentrations in PRD (42.6±33.7 ppb) and in NCP (81.7±57.0 ppb) (Figure S22) would**
**cause $RO_2$ radicals from organic compounds mainly reacting with NO (Bahreini et al.,**
**2009)."**

[Figure]

Figure S22. Time series of NO$_x$ during the PRD (a) and the NCP (b) campaigns, respectively.

13. Lines 299-300: The sentence reads as if isoprene and monoterpenes are emitted in vehicle exhaust. Please correct.

Response: We deleted "including vehicle exhausts and biomass combustions in this region".

The sentence is modified to **"As mentioned above, we attribute these isoprene and**

**monoterpenes to anthropogenic emissions in this region."**

14. Line 312: Can you describe how this is 'independent'?

Response: We replaced the "independent" with "another".

15. Line 332: The previous literature that has examined alkanes in more detail (Gentner et al.,

2012; Zhao et al., 2015,2016; Drozd et al., 2019) have found a strong contribution of cyclic alkanes, especially in vehicular exhaust. Were these specifically measured/estimated in this work and can the authors comment on their role in contributing to SOA formation?

Response: In this study, we didn't estimate the contribution of cyclic alkanes to SOA formation.

We are trying to establish the quantification method of these cyclic alkanes.

Reference:

Apel, E. C., Riemer, D. D., Hills, A., Baugh, W., Orlando, J., Faloona, I., Tan, D., Brune, W.,

Lamb, B., Westberg, H., Carroll, M. A., Thornberry, T., and Geron, C. D.: Measurement and interpretation of isoprene fluxes and isoprene, methacrolein, and methyl vinyl ketone mixing ratios at the PROPHET site during the 1998 Intensive, Journal of Geophysical Research:

Atmospheres, 107, ACH 7-1-ACH 7-15, 10.1029/2000JD000225, 2002.

Bahreini, R., B. Ervens, A. M. Middlebrook, C. Warneke, J. A. de Gouw, P. F. DeCarlo, J. L.

Jimenez, C. A. Brock, J. A. Neuman, T. B. Ryerson, H. Stark, E. Atlas, J. Brioude, A. Fried, J.

S. Holloway, J. Peischl, D. Richter, J. Walega, P. Weibring, A. G. Wollny and F. C. Fehsenfeld .

Organic aerosol formation in urban and industrial plumes near Houston and Dallas, Texas.

Journal of Geophysical Research: Atmospheres 114(D7), 10.1029/2008JD011493, 2009.

Cappa, C. D. and Wilson, K. R.: Multi-generation gas-phase oxidation, equilibrium partitioning, and the formation and evolution of secondary organic aerosol, Atmos. Chem. Phys., 12, 9505-

9528,doi:10.5194/acp-12-9505-2012, 2012.

Gentner, D. R., Isaacman, G., Worton, D. R., Chan, A. W. H., Dallmann, T. R., Davis, L., Liu,

S., Day, D.A., Russell, L. M., Wilson, K. R., Weber, R., Guha, A., Harley, R. A., and Goldstein,

A. H.: Elucidating secondary organic aerosol from diesel and gasoline vehicles through detailed characterization of organic carbon emissions, Proceedings of the National Academy of Sciences of the United States of America, 109, 18318-18323, 10.1073/pnas.1212272109,

2012.

Roberts, J. M., Fehsenfeld, F. C., Liu, S. C., Bollinger, M. J., Hahn, C., Albritton, D. L., and

Sievers, R. E.: Measurements of aromatic hydrocarbon ratios and NOx concentrations in the rural troposphere: Observation of air mass photochemical aging and NOx removal,

Atmospheric Environment (1967), 18, 2421-2432, https://doi.org/10.1016/0004-

6981(84)90012-X, 1984.

Parrish, D. D., A. Stohl, C. Forster, E. L. Atlas, D. R. Blake, P. D. Goldan, W. C. Kuster and

J. A. de Gouw. Effects of mixing on evolution of hydrocarbon ratios in the troposphere. Journal of Geophysical Research: Atmospheres 112(D10), 2007.

Roberts, J., Marchewka, M., Bertman, S., Goldan, P., Kuster, W., de Gouw, J., warneke, C.,

Williams, E., Lerner, B., Murphy, P., Apel, E., and Fehsenfeld, F.: Analysis of the isoprene chemistry observed during the New England Air Quality Study (NEAQS) 2002 Intensive

Experiment, Journal of Geophysical Research-Atmospheres, 111, D23S12,
10.1029/2006JD007570, 2006.

Zhang, X., C. D. Cappa, S. H. Jathar, R. C. McVay, J. J. Ensberg, M. J. Kleeman and J. H.
Seinfeld. Influence of vapor wall loss in laboratory chambers on yields of secondary organic
aerosol. Proceedings of the National Academy of Sciences of the United States of America,
111(16): 5802-5807, 10.1073/pnas.1404727111, 2014.

---

## Referee Report (RR1)

Reviewer comments on acp-2020-145 revision

This manuscript, "Measurements of higher alkanes using NO+ chemical ionization in PTR-ToF-MS: important contributions of higher alkanes to secondary organic aerosols in China," is a revision of a manuscript I previously reviewed. In it the authors study the contribution of high-molecular-weight gas-phase alkanes to SOA formation, specifically at two sites in China. The authors seem to have addressed most of my original concerns, though I note below a few things that are still of general concern and could be addressed better. I think the manuscript is generally suitable for publication, but would encourage the authors to consider some of these comments and suggests.

General comments:

1) The authors provide much improved discussion of the sampling inlet design, and present data (Figure S11) demonstrating that the 8 meter length of tubing does not impact C8-C15 alkanes. I would note, however, that there are some limitations to their tests that leave me only partly convinced. For example, the C15 cal curve is not completely linear: 2 ppb gives a response below 200 ncps though 1 ppb gives a response of 100 ncps, but the same non-linearity is not observed for C10. There is also a clear time lag in the 8-meter tube that grows with carbon number, though it remains fast. Furthermore, the true sampling set up has many valves in line between the ambient sample and the instrument (6, by my count), which could substantially increase the surfaces and thus the time lag, and this effect is not considered in the tubing test.

Unfortunately, only up to C15 is tested, but these issues are likely to get much worse for up to C21. This effect can be modeled using the work of Pagonis et al. (Atmos. Meas. Tech., 10, 4687–4696, 2017). I have included a figure of this model belo for the sampling system here (8 meter tube with 5 lpm flow), but roughly, C10 and smaller alkanes have essentially no lag, and up to C15 have lags on the order of a minute or so. Notably, for larger alkanes, time delays increase, with C21 having a lag of hours; while this is likely a worst case, I would note that the estimates for C8-15 are more or less in agreemeent with Figure S11, so the model seems to be working to some degree. In addition to potential losses, this might have significant impacts on measured diurnals, etc.

[Figure]

Because much of the results of this work are not dependent on the diurnals or time-dependent measurements of higher-alkanes, I don't know that this issue is fatal. However, if the authors intend to continue these types of measurements, they will need to convince themselves (and reviewers) that their measurements of these lower-volatility gases are reliable. One option would be to show that diurnals for lower-volatility alkanes exhibit similar time dependence as higher-volatility alkanes, another would be to measure the time constant for step-function changes in concentrations of low-volatility alkanes even if quantitative concentrations can't be reliably generated.

2) The new detail and discussion around OH expsoure calculations are much improved. One suggestion I would make, though, is to change the language a little around this topic. I think many of us think of "OH exposure" as a feature of an air mass, so it is a bit confusing to talk about different OH exposure for different components, e.g., aromatics vs biogenics. The authors discussion in their response to reviewers of the reasons for this (differences in timing of the different emission sources) is helpful in understanding this. I think it might help to think of these calculations as "photochemical age", which could be described as the time between emission and detection - it might be more intuitive that this is different between components, and some inclusion of this discussion might help with interpretation of Figure S16.

Specific comments:

Line 315. It is not really true that branched alkanes react faster than normal alkanes with OH in the gas phase. Though Isaacman et al. showed this was true in the particle phase (Environ. Sci. Technol. 2012, 46, 10632–10640), structure activity relationshops do not predict the same effect in the gas-phase. Roughly speaking, branching increases the number of tertiary carbons, but concomitantly increases the number of primary carbons - these effects balance out at the estimated OH rate constant is mostly unchanged. For example, Kwok and Atkinson methods estimate kOH=2.44e11 for 2,6,10,14-tetramethylpentadecane ("pristane") and kOH=2.38e11 for n-nonadecane, a difference of less than 3%, well within uncertainty. The lower estimated yields for branched alkanes are consequently expected to "win out" in these calculations, since the effect of branching on yields is likely more substantial (Gentner et al. estimated branched yields were roughly half that of normal alkanes).

Figure 5. In their response to reviewers, the authors state that Worton et al. is not included because that work focused on particle-phase alkanes. However, I believe the same fact is true of the Chan et al. work, so I'm not sure that is a reasonable argument.

---

## Author Response (AR2)

**Response to anonymous referee #1**

Reviewer comments on acp-2020-145 revision

This manuscript, "Measurements of higher alkanes using $NO^+$ chemical ionization in PTR-ToF-MS: important contributions of higher alkanes to secondary organic aerosols in China," is a revision of a manuscript I previously reviewed. In it the authors study the contribution of high-molecular-weight gasphase alkanes to SOA formation, specifically at two sites in China. The authors seem to have addressed most of my original concerns, though I note below a few things that are still of general concern and could be addressed better. I think the manuscript is generally suitable for publication, but would encourage the authors to consider some of these comments and suggests.

Response: Thank you very much for your valuable comments and suggestions on our revised manuscript. These comments are all helpful for improving our article. All the authors have seriously discussed about all these comments. According to your comments, we have tried best to modify our manuscript to meet with the requirements for the publication in this journal. Point-by-point responses to the comments are listed below.

General comments:

1) The authors provide much improved discussion of the sampling inlet design, and present data (Figure S11) demonstrating that the 8 meter length of tubing does not impact C8-C15 alkanes. I would note, however, that there are some limitations to their tests that leave me only partly convinced. For example, the C15 cal curve is not completely linear: 2 ppb gives a response below 200 ncps though 1 ppb gives a response of 100 ncps, but the same non-linearity is not observed for C10. There is also a clear time lag in the 8-meter tube that grows with carbon number, though it remains fast. Furthermore, the true sampling set up has many valves in line between the ambient sample and the instrument (6, by my count), which could substantially increase the surfaces and thus the time lag, and this effect is not considered in the tubing test.

Unfortunately, only up to C15 is tested, but these issues are likely to get much worse for up to C21. This effect can be modeled using the work of Pagonis et al. (Atmos. Meas. Tech., 10, 4687-4696, 2017). I have included a figure of this model belo for the sampling system here (8 meter tube with 5 lpm flow), but roughly, C10 and smaller alkanes have essentially no lag, and up to C15 have lags on the order of a minute or so.

Notably, for larger alkanes, time delays increase, with C21 having a lag of hours; while this is likely a worst case, I would note that the estimates for C8-15 are more or less in agreemeent with Figure S11, so the model seems to be working to some degree. In addition to potential losses, this might have significant impacts on measured diurnals, etc.

[Figure]

Because much of the results of this work are not dependent on the diurnals or time-dependent measurements of higher-alkanes, I don't know that this issue is fatal. However, if the authors intend to continue these types of measurements, they will need to convince themselves (and reviewers) that their measurements of these lower-volatility gases are reliable. One option would be to show that diurnals for lower-volatility alkanes exhibit similar time dependence as higher-volatility alkanes, another would be to measure the time constant for step-function changes in concentrations of low-volatility alkanes even if quantitative concentrations can't be reliably generated.

Response: Thank you very much for your comment and nice suggestion. We re-calculated the delay time of higher alkanes, which is determined as the time it takes for the signal to drop to 10% of its initial value casued by the step-function change in sample concentration (Pagonis et al., 2017). The results are from measurements during the field campaigns described in this study and also some other measurements that were conducted by our group, including measurements of emission sources and tubing losses test in laboratory. The delay times for higher alkanes are summarized Figure S10. It is found that delay times for various alkanes are in a range of few seconds to few minutes, among of which, higher-volatility alkanes (C8-C15) are better than 1 min and lower-volatility alkanes (C16-C21) are relatively long reaching several minutes. These results suggest that alkanes with higher carbon number, especially C20 and C21 might be

influenced by the tubing delay effect, as PTR-ToF-MS measured higher alkanes 10 minutes for ambient air and 3 minutes for background. We genearally observe longer delay time with tubing (5 m or 8 m) than only instrument, consistent with the results in (Pagonis et al., 2017). However, the determined delay time for alkanes with carbon number larger than 18 is signifciantly lower than the modelled delay time by the reviewer using the model in (Pagonis et al., 2017).

As suggested by the reviewer, we also included the diurnal variations of all the measured alkanes (C8-C21) in this study in SI of the revised manuscript. As shown in the attached figure, the lower-volatility alkanes exhibit very similar time dependence as higher-volatility alkanes during both campaigns in PRD and NCP. These results also imply that the tubing effects should not sinificantly affect on the temperoal variations of higher alkanes in this study.

As discussed above, the determination of tubing delay is really important for accurate measurements of higher alkanes and other intermediate volatile species. We agree with the reviewer and also strongly suggest to characterize the tubing used for this type of measurements. As suggested in (Pagonis et al., 2017) and other campanion papers (Liu et al., 2019), shorter inlet and higher flow through the inlet are better practice for minimizing the tubing delay effect.

We extended the discussion of delay time on lines 186-198 of page 9-10 in the revised manuscript as follows:

**"Delay time is calculated as the time it takes for the signal to drop to 10% of its initial value caused by the step-function change in sample concentration (Pagonis et al., 2017). The delay times of higher alkanes for the field measurements in this study and some other measurements (e.g. emission source measurements and tubing losses test in the laboratory are summarized in Figure S10. It is found that delay times for various alkanes are in a range of few seconds to few minutes, among of which, higher-volatility alkanes (C8-C15) are better than 1 min and lower-volatility alkanes (C16-C21) are relatively long reaching several minutes. These results suggest that alkanes with higher carbon number, especially C20 and C21 might be influenced by the tubing delay effect during the measurements. However, as shown later in section 3.1, the lower-volatility alkanes exhibit very similar diurnal variations as higher-volotility alkanes during both campaigns in PRD and NCP, implying that the tubing effects should not**

**sinificantly affect on temperoal variations of higher alkanes reported in this study.”**

[Figure]

Figure S10. Delay times of higher alkanes for the field campaigns, emission source measurements and tubing losses test in the laboratory.

[Figure]

Figure S17. Similar diurnal profiles of C8-C21 alkanes during campaigns in PRD (a, b) and NCP (c, d).

Reference:

Pagonis, D., J. E. Krechmer, J. de Gouw, J. L. Jimenez and P. J. Ziemann: Effects of gas-wall partitioning in Teflon tubing and instrumentation on time-resolved measurements of gas-phase organic compounds. Atmos. Meas. Tech. 10(12): 4687-4696, 10.5194/amt-10-4687-2017, 2017.

Liu, X., Deming, B., Pagonis, D., Day, D. A., Palm, B. B., Talukdar, R., Roberts, J. M., Veres, P. R., Krechmer, J. E., Thornton, J. A., de Gouw, J. A., Ziemann, P. J., and Jimenez, J. L.: Effects of gas-wall interactions on measurements of semivolatile compounds and small polar molecules, Atmospheric Measurement Techniques, 12, 3137-3149, 10.5194/amt-12-3137-2019, 2019.

2) The new detail and discussion around OH exposure calculations are much improved. One suggestion I would make, though, is to change the language a little around this topic. I think many of us think of "OH exposure" as a feature of an air mass, so it is a bit confusing to talk about different OH exposure for different components, e.g., aromatics vs biogenics. The authors discussion in their response to reviewers of the reasons for this (differences in timing of the different emission sources) is helpful in understanding this. I think it might help to think of these calculations as "photochemical age", which could be described as the time between emission and detection - it might be more intuitive that this is different between components, and some inclusion of this discussion might help with interpretation of Figure S16.

Response: Thank you very much for your nice suggestion. In the main body of this study, we use the following formula to calculate the SOA formation for a given compound:

$$[SOA_i]_t = [VOC_i]_t \times (e^{k_{VOC_i} \times ([OH] \times \Delta t)} - 1) \times Yield_i$$

In the above formula, $\Delta t$ represents the photochemical age, $[OH]$ represents the OH concentration. In this study, we calculate the $[OH] \times \Delta t$, which was considered as OH exposure in some studies (Jimenez et al., 2009).

We added sentences on lines 297-299 of page 13 in the revised manuscript to mention the "OH exposure" and "photochemical age" clearly as follows:

"**$[OH]$ is the OH concentration (molecules cm$^{-3}$), $\Delta t$ is the photochemical age. In this study, we calculate the $[OH] \times \Delta t$ (molecules cm$^{-3}$ s), which was considered as OH exposure in some studies (Jimenez et al., 2009).**"

Reference:

Jimenez, J. L., et al. (2009), Evolution of Organic Aerosols in the Atmosphere, Science, 326(5959), 1525-1529.

Specific comments:

Line 315. It is not really true that branched alkanes react faster than normal alkanes with OH in the gas phase. Though Isaacman et al. showed this was true in the particle phase (Environ. Sci. Technol. 2012, 46, 10632-10640), structure activity relationships do not predict the same effect in the gas-phase. Roughly speaking, branching increases the number of tertiary carbons, but concomitantly increases the number of primary carbons - these effects balance out at the estimated OH rate constant is mostly unchanged. For example, Kwok and Atkinson methods estimate kOH=2.44e11 for 2,6,10,14-tetramethylpentadecane ("pristane") and kOH=2.38e11 for n-nonadecane, a difference of less than 3%, well within uncertainty. The lower estimated yields for branched alkanes are consequently expected to "win out" in these calculations, since the effect of branching on yields is likely more substantial (Gentner et al. estimated branched yields were roughly half that of normal alkanes).

Response: Thank you very much for your correction. We modified the sentences on lines 329-332 of page 14 in the revised manuscript as follows:

"**Considering the SOA yields of branched alkanes are lower than *n*-alkanes, which is depending on chemical structures of the carbon backbone (Lim and Ziemann, 2009;Tkacik et al., 2012;Loza et al., 2014), the estimation of SOA from alkanes in this study might be a little overestimated.**"

Reference:

Lim, Y. B., and Ziemann, P. J.: Effects of Molecular Structure on Aerosol Yields from OH Radical-Initiated Reactions of Linear, Branched, and Cyclic Alkanes in the Presence of NOx, Environmental Science & Technology, 43, 2328-2334, 10.1021/es803389s, 2009.

Loza, C. L., Craven, J. S., Yee, L. D., Coggon, M. M., Schwantes, R. H., Shiraiwa, M., Zhang, X., Schilling, K. A., Ng, N. L., Canagaratna, M. R., Ziemann, P. J., Flagan, R. C., and Seinfeld, J. H.: Secondary organic aerosol yields of 12-carbon alkanes, Atmospheric Chemistry and Physics, 14, 1423-1439, 10.5194/acp-14-1423-2014, 2014.

Tkacik, D. S., Presto, A. A., Donahue, N. M., and Robinson, A. L.: Secondary Organic Aerosol Formation from Intermediate-Volatility Organic Compounds: Cyclic, Linear, and Branched Alkanes, Environmental Science & Technology, 46, 8773-8781, 10.1021/es301112c, 2012.

Figure 6. In their response to reviewers, the authors state that Worton et al. is not included because that work focused on particle-phase alkanes. However, I believe the same fact is true of the Chan et al. work, so I'm not sure that is a reasonable argument.

Response: Thank you very much for your correction. Yes, the higher alkanes data reported in Worton et al., 2014 and Chan et al., 2013 are both in particle phase. Hence, we also included the work from Worton et al., 2014 in the revised figure. We want to show the importance of contribution of branched isomers to higher alkanes concentrations at each carbon number whatever in gas phase and particle phase.

The Figure 6 is modified as follows:

[Figure]

Figure 6. Fractions of *n*-alkanes in higher alkanes with same formulas in gas phase (hollow dots) and particle phase (solid dots) derived from this study, ambient air in Los

Angeles, Bakersfield, Caldecott Tunnel and in vehicle exhausts (Chan et al., 2013; Gentner et al., 2012; Worton et al., 2014).

The Table S2 are modified as follows:

Table S2. Fractions of *n*-alkanes in higher alkanes with same formulas derived from this study, ambient air in Los Angeles, Bakersfield, Caldecott Tunnel and in vehicle exhausts.

| Carbon Number | Fraction of *n*-alkanes in higher alkanes with same formulas | | | | | |
|---|---|---|---|---|---|---|
| | PRD[a] | Los Angeles[b] | Bakersfield[b] | Caldecott Tunnel[c] | Diese exhaust[d] | Liquid gasoline exhaust[d] |
| 8 | 10.82 | / | / | / | 37.04 | 5.39 |
| 9 | 21.48 | / | / | / | 51.22 | 7.71 |
| 10 | 17.56 | / | / | / | 23.81 | 8.81 |
| 11 | 17.81 | / | / | / | 20.91 | 10.88 |
| 12 | / | / | / | / | 22.54 | 29.82 |
| 13 | / | / | / | / | 21.98 | / |
| 14 | / | / | / | / | 19.84 | 5.41 |
| 15 | / | / | / | / | 22.86 | 40 |
| 16 | / | / | / | / | 25.44 | / |
| 17 | / | / | / | / | 32.16 | / |
| 18 | / | / | / | / | 28.57 | / |
| 19 | / | / | / | / | 20.83 | / |
| 20 | / | 34.78 | 24.85 | / | 20.87 | / |
| 21 | / | 53.16 | 12.47 | / | 24.82 | / |
| 22 | / | 42.85 | 9.11 | / | 25.51 | / |
| 23 | / | 40.24 | 8.14 | 58.82 | 21.05 | / |
| 24 | / | 21.85 | 6.92 | 34.62 | 24.44 | / |
| 25 | / | 27.17 | 27.87 | 32.35 | 60 | / |
| 26 | / | / | / | 25 | / | / |
| 27 | / | / | / | 27.03 | / | / |
| 28 | / | / | / | 38.64 | / | / |
| 29 | / | / | / | 29.63 | / | / |
| 30 | / | / | / | 23.53 | / | / |

[a]:This work; [b]: Chan et al. (2013); [c]: Worton et al. (2014); [d]: Gentner et al. (2012)

We also modified the sentences on lines 253-255 of page 11 as follows:

"**We found the fractions are in the range of 11%-21% for carbon number of 8-11, which are comparable with results of ambient air in California, tunnel test and vehicle exhausts (Figure 6 and Table S2) (Chan et al., 2013; Worton et al., 2014; Gentner et al., 2012)**."

Reference:

Chan, A. W. H., Isaacman, G., Wilson, K. R., Worton, D. R., Ruehl, C. R., Nah, T., Gentner, D. R., Dallmann, T. R., Kirchstetter, T. W., Harley, R. A., Gilman, J. B., Kuster, W. C., deGouw, J. A., Offenberg, J. H., Kleindienst, T. E., Lin, Y. H., Rubitschun, C. L., Surratt, J. D., Hayes, P. L., Jimenez, J. L., and Goldstein, A. H.: Detailed chemical characterization of unresolved complex mixtures in atmospheric organics: Insights into emission sources, atmospheric processing, and secondary organic aerosol formation, Journal of Geophysical Research-Atmospheres, 118, 6783-6796, 10.1002/jgrd.50533, 2013.

Gentner, D. R., Isaacman, G., Worton, D. R., Chan, A. W. H., Dallmann, T. R., Davis, L., Liu, S., Day, D. A., Russell, L. M., Wilson, K. R., Weber, R., Guha, A., Harley, R. A., and Goldstein, A. H.: Elucidating secondary organic aerosol from diesel and gasoline vehicles through detailed characterization of organic carbon emissions, Proceedings of the National Academy of Sciences of the United States of America, 109, 18318-18323, 10.1073/pnas.1212272109, 2012.

Worton, D. R., G. Isaacman, D. R. Gentner, T. R. Dallmann, A. W. H. Chan, C. Ruehl, T. W. Kirchstetter, K. R. Wilson, R. A. Harley and A. H. Goldstein: Lubricating Oil Dominates Primary Organic Aerosol Emissions from Motor Vehicles. Environmental Science & Technology, 48(7): 3698-3706, 10.1021/es405375j, 2014.

**Response to anonymous referee #2**

The authors addressed all of my comments from the first round of review, and seem to have addressed the comments of the other reviewers as well. I have a few minor comments below.

Response: Thank you very much for your valuable comments and suggestions on our revised manuscript. These comments are all helpful for improving our article. All the authors have seriously discussed about all these comments. According to your comments, we have tried best to modify our manuscript to meet with the requirements for the publication in this journal. Point-by-point responses to the comments are listed below.

The only major issue with the revised manuscript is in Figure 8. I generally like this figure, but I don't understand the shaded parts of the bars labelled "consumed", as it is not explained in the text. Please clarify. Also, why are there two different lines for SOA yield in part (a)? I think part (b), which shows that the peak in alkane SOA production occurs around C15, is a great result that shows the importance of IVOCs on SOA.

Response: We clarified the "consumed" on lines 798-800 of page 41 in the revised manuscript as follows:

"**The consumed concentrations represent the chemical losses of higher alkanes, which are calculated by using the estimated SOA from each alkane dividing the corresponding SOA yields.**"

We also modified the sentences on lines 342-345 of page 15 as follows:

"**Larger alkanes are relatively more reactive than lighter ones, which results in larger proportions of calculated concentrations that have been chemically consumed in the atmosphere (the concentrations labelled "consumed" in Figure 8(a)).**"

The two different lines for SOA yields represent the yields that we used in PRD and NCP campaigns, respectively. SOA yield data used in this study for higher alkanes were summarized from reported values in the literature, with the consideration of the influence of organic aerosol concentration (Figure S20) to SOA yields (Donahue et al., 2006) (Figure S21). Since the organic aerosol concentrations are quite different in these two campaigns, the SOA yields of high alkanes are correspondingly different.

We added sentences on lines 348-350 of page 15 in the revised manuscript to mention the peaks of SOA production from alkanes as follows:

**"The peaks in alkanes SOA productions occur around C15 in both campaigns of PRD and NCP, which is a great result that shows the importance of IVOCs on SOA."**

One comment on Figure 5c - the diurnal pattern of total alkanes from NO+-PTR has a deeper afternoon trough than the n-alkanes measured by GC-MS. This suggests that the n-alkanes cannot be used as tracers for the total alkane signal at each carbon number.

Response: Thank you very much for your nice suggestion. We added sentences in the revised manuscript to mention this result on lines 245-248 of page 11 as follows: **"However, the diurnal patterns of total alkanes from $NO^+$ PTR-ToF-MS have a deeper afternoon trough than the *n*-alkanes measured by GC-MS, implying that *n*-alkanes may have different temporal variations compared with those of total alkanes."**

We also discussed the importance of $NO^+$ PTR-ToF-MS in measuring higher alkanes on lines 256-259 of page 11 as follows:

**"These results indicate the importance of branched alkanes in concentrations of higher alkanes and their potential contributions to SOA formation. It also has strong implication for the merits of $NO^+$ PTR-ToF-MS in measuring sum of the alkanes with the same formula for estimation of SOA contributions, as discussed later."**

Figure 9 - the authors do not seem to comment on the poor SOA mass closure. This of course could be due to several factors (yield estimates, unmeasured species, etc). I would like to see the mass closure issue commented on before publication. I'm also interested to know how much better the mass closure is when using the $NO^+$ data in addition to the traditional PTR-MS data (e.g., what is the predicted SOA formation if only the H+ PTR-MS data was available?).

Response: Thank you very much for your comment and nice suggestion. We modified the sentences on lines 371-377 of page 16 in the revised manuscript to mention the SOA mass closure as follows:

**"The low explained percentages of SOA formations in both of PRD and NCP (Figure 9(c, d)) imply that some other SOA precursors (e.g. alkylcyclohexanes, alkylbenzenes, cyclic and polycyclic aliphatic materials) (Zhao et al., 2015; Drozd et al., 2019) or formation pathways (e.g. aqueous reactions) (Kuang et al., 2020) are contributing significantly to SOA formation. Compared to a previous study in northern China (Yuan et al., 2013), the missing gap of SOA formation declined after explicitly considering higher alkanes and naphthalenes in SOA production."**

Reference:

Drozd, G. T., Zhao, Y., Saliba, G., Frodin, B., Maddox, C., Oliver Chang, M. C., Maldonado, H., Sardar, S., Weber, R. J., Robinson, A. L., and Goldstein, A. H.: Detailed Speciation of Intermediate Volatility and Semivolatile Organic Compound Emissions from Gasoline Vehicles: Effects of Cold-Starts and Implications for Secondary Organic Aerosol Formation, Environ. Sci. Technol., 53, 1706-1714, https://doi.org/10.1021/acs.est.8b05600, 2019.

Kuang, Y., He, Y., Xu, W., Yuan, B., Zhang, G., Ma, Z., Wu, C., Wang, C., Wang, S., Zhang, S., Tao, J., Ma, N., Su, H., Cheng, Y., Shao, M., and Sun, Y.: Photochemical Aqueous-Phase Reactions Induce Rapid Daytime Formation of Oxygenated Organic Aerosol on the North China Plain, Environmental Science & Technology, 10.1021/acs.est.9b06836, 2020.

Zhao, Y., Nguyen, N. T., Presto, A. A., Hennigan, C. J., May, A. A., and Robinson, A. L.: Intermediate Volatility Organic Compound Emissions from On-Road Diesel Vehicles: Chemical Composition, Emission Factors, and Estimated Secondary Organic Aerosol Production, Environ Sci Technol, 49, 11516-11526, 10.1021/acs.est.5b02841, 2015.

Line 62 - define NMHCs

Response: We defined NMHCs on lines 62-63 of page 3 in the revised manuscript as follows: "**nonmethane hydrocarbons (NMHCs)**".

Lines 136-138 - I do not understand what the authors mean by "with signals either the largest or among the largest ions at these nominal masses, which help to achieve high precision for determined signals of higher alkanes from high-resolution peak fitting ."

Response: In this study, the measured mass spectra from NO⁺ PTR-ToF-MS was analysed using Tofware software (Tofwerk AG) for high-resolution peak-fitting. Higher alkanes were detected through hydride abstraction by NO⁺ forming mass (m-1) ions (m is the molecular mass) (Koss et al., 2016;Inomata et al., 2013). As shown in the attached Figure 1, the product ions (m-1) of higher alkanes have relatively high signals at these nominal masses, for example, the signal of $C_{20}H_{41}^+$ produced by eicosane is the highest at m/z 281, which helps in getting precise peak-fitting results (Cubison and Jimenez, 2015;Corbin et al., 2015). That is to say, if the signals of product ions (m-1) of higher alkanes are too small, then the peak-fitting results may have large uncertainties.

[Figure]

**Figure 1.** High-resolution (HR) peak-fitting to the averaged mass spectra on a typical day (12 October 2018) for m/z 169 **(a)**, m/z 211 **(b)** and m/z 281 **(c)**, at which masses

produced by dodecane ($C_{12}H_{25}^+$), pentadecane ($C_{15}H_{31}^+$) and eicosane ($C_{20}H_{41}^+$) in $NO^+$ PTR-ToF-MS.

Reference:

Corbin, J. C., Othman, A., D. Allan, J., R. Worsnop, D., D. Haskins, J., Sierau, B., Lohmann, U., and A. Mensah, A.: Peak-fitting and integration imprecision in the Aerodyne aerosol mass spectrometer: effects of mass accuracy on location-constrained fits, Atmos. Meas. Tech., 8, 4615-4636, 10.5194/amt-8-4615-2015, 2015.

Cubison, M. J., and Jimenez, J. L.: Statistical precision of the intensities retrieved from constrained fitting of overlapping peaks in high-resolution mass spectra, Atmos. Meas. Tech., 8, 2333-2345, 10.5194/amt-8-2333-2015, 2015.